# WHEN SHIFT HAPPENS - CONFOUNDING IS TO BLAME

**Abbavaram Gowtham Reddy, Celia Rubio-Madrigal, Rebekka Burkholz,**[*] **Krikamol Muandet**[*]
CISPA Helmholtz Center for Information Security, Saarbrücken, Germany

## ABSTRACT

Distribution shifts introduce uncertainty that undermines the robustness and generalization capabilities of machine learning models. While conventional wisdom suggests that learning causal-invariant representations enhances robustness to such shifts, recent empirical studies present a counterintuitive finding: (i) empirical risk minimization (ERM) can rival or even outperform state-of-the-art out-of-distribution (OOD) generalization methods, and (ii) OOD generalization performance improves when all available covariates, including non-causal ones, are utilized. We present theoretical and empirical explanations that attribute this phenomenon to hidden confounding. Shifts in hidden confounding induce changes in data distributions that violate assumptions commonly made by existing approaches. Under such conditions, we prove that generalization requires learning environment-specific relationships, rather than relying solely on invariant ones. Furthermore, we explain why models augmented with non-causal but informative covariates can mitigate the challenges posed by hidden confounding shifts. These findings offer new theoretical insights and practical guidance, serving as a roadmap for future research on OOD generalization and principled covariate-selection strategies.

## 1 INTRODUCTION

Generalization—the ability to draw reliable conclusions about unseen data based on observed data—is central to numerous scientific fields. In medicine and the social sciences, it is embodied as *external validity*, ensuring findings from one population are applicable to a different population (Campbell & Stanley, 2015); in ecology, it supports *space-for-time substitutions*, where spatial variation is used as a proxy for temporal change to infer long-term ecological patterns (Pickett, 1989); and in engineering, it drives *robust control*, where models must maintain performance in the presence of unmodeled disturbances (Khalil et al., 1996). In recent years, machine learning has become a powerful tool for learning generalizable models (Zhou et al., 2022; Wang et al., 2022; Liu et al., 2021).

To generalize well, machine learning models must be robust to distribution shifts between training and test data. For example, a model trained to predict *food stamp recipiency* based on *household attributes* in one region should be capable of adapting and performing well when deployed in another region. A model is said to achieve *out-of-distribution (OOD) generalization* when it maintains its performance on both *in-distribution (ID)* data (from which training data are sampled) and OOD test data. Over the last decade, various families of methods such as domain generalization (Muandet et al., 2013; Heinze-Deml et al., 2018; Zhao et al., 2022; Singh et al., 2024), domain adaptation (Zhao et al., 2019; Long et al., 2018; Xu et al., 2020; Sun & Saenko, 2016a), and robust learning (Levy et al., 2020; Sagawa et al., 2019) have been proposed to achieve provable OOD generalization under specific assumptions. However, under careful model selection, models based on standard *empirical risk minimization (ERM)* (Vapnik, 1999) often achieve competitive OOD generalization performance across a range of real-world applications (Gulrajani & Lopez-Paz, 2021; Krueger et al., 2021; Liu et al., 2023; Nastl & Hardt, 2024; Rosenfeld et al., 2022; Vedantam et al., 2021). Moreover, a recent empirical study (Nastl & Hardt, 2024) on 16 real-world tabular datasets has concluded that incorporating all available covariates when predicting the outcome, regardless of whether they directly affect the outcome, can improve OOD generalization performance. These findings challenge prevailing assumptions in the field and motivate a deeper investigation into the mechanisms underlying OOD generalization.

Distribution shifts are commonly observed when data originate from different environments (Schölkopf, 2022). For example, *food stamp recipiency* may differ across states, because each

---

[*]These authors share senior authorship.

state operates under different eligibility rules, leading to distinct, environment-specific distributions. In such data, certain statistical relationships between covariates and the outcome may stay consistent across all environments, referred to as *invariant relationships* (Arjovsky et al., 2019). Identifying and learning such invariant relationships ensures OOD generalization (Arjovsky et al., 2019; Peters et al., 2016; Rojas-Carulla et al., 2018; Muandet et al., 2013; Quinzan et al., 2024). A special kind of invariance is known as *causal invariance*, where the causal relationships in the data stay invariant across environments (Schölkopf, 2022; Peters et al., 2016; Arjovsky et al., 2019). However, in many real-world settings, not all relevant covariates required for predicting an outcome are observed, due to limitations in data collection, privacy constraints, or measurement errors (Carroll et al., 2006; Louizos et al., 2017; Dwork et al., 2014; Little & Rubin, 2019). Their absence disrupts the invariant relationships needed for models to achieve generalization. This issue is further compounded when the unobserved variables are confounders that influence both the observed covariates and the outcome. In practice, such hidden confounders are pervasive, and shifts in their distributions correspond to distinct environments. Ignoring these shifts not only undermines generalization performance (Landeiro & Culotta, 2018; Alabdulmohsin et al., 2023; Tsai et al., 2024; Prashant et al., 2025), but can also lead to learning incorrect relationships between observed covariates and the outcome.

*Despite its importance and the recent progress on achieving generalization under hidden confounding shifts* (Alabdulmohsin et al., 2023; Tsai et al., 2024; Prashant et al., 2025), *how hidden confounding shift affects generalization and how informative covariates help in generalization remains poorly understood. Our work aims to bridge this gap by providing theoretical and empirical explanations.*

Our contributions are as follows:

- We motivate the need for studying confounding shift in OOD generalization from a causal perspective (§ 3) and explain why adding informative, non-causal covariates can improve performance.

- While invariant representations alone, while sufficient, are challenging to achieve under the hidden confounding shift. We show that maximizing predictive information between model predictions and true outcomes demands explicitly learning environment-specific relationships. (§ 4.1, § 4.2).

- We demonstrate that variables informative of either outcome or hidden confounders help in improving predictive information between model predictions and true outcomes. This explains the importance of principled covariate selection in the presence of hidden confounding shift (§ 4.3).

- Our experiments on both real-world and synthetic datasets provide evidence that (i) hidden confounding is prevalent in real-world tabular benchmark data, (ii) learning environment-specific relationships correlates positively with ID and OOD test accuracy, and (iii) incorporating informative, non-causal covariates improves generalization (§ 5).

## 2 RELATED WORK

Out-of-distribution (OOD) generalization encompasses various facets, with notable examples including domain generalization (Zhou et al., 2022; Wang et al., 2022), domain adaptation (Zhao et al., 2019; Sun & Saenko, 2016b; Long et al., 2018; Xu et al., 2020), robust learning (Levy et al., 2020; Sagawa et al., 2019), federated learning (Li et al., 2023), and OOD detection (Lee et al., 2018; Hendrycks & Gimpel, 2017). Domain adaptation assumes access to unlabeled data from the test set, whereas domain generalization relies on environment labels during training. Federated learning adopts a collaborative learning framework, tackling constraints such as communication efficiency and privacy when data originates from multiple environments. OOD detection focuses on identifying samples that differ significantly from the training distribution. A common goal of many of these methods is to learn invariant relationships. However, recent work suggests that additional inductive biases beyond invariance are required for improved generalization (Lin et al., 2022; Schrouff et al., 2022; Ye et al., 2021). Ye et al. (2021) argue that invariance of features is necessary but not sufficient for generalization and discuss the importance of *informativeness* of features for generalization. We explain how informativeness plays a crucial role in generalization under hidden confounding shifts.

**Proxy variable adjustment:** When confounding variables are observed during training and unobserved at test time, Landeiro & Culotta (2018) propose adjusting for confounding shifts to improve classifier performance. Building on proxy-based adjustment methods for causal effect estimation (Miao et al., 2018; Kuroki & Pearl, 2014), recent domain adaptation methods rephrase the problem of unknown distribution shifts as a causal effect identification problem (Alabdulmohsin

et al., 2023; Tsai et al., 2024). Recently, OOD generalization under hidden confounding shift has been considered under the assumption of overlapping confounder support (Prashant et al., 2025). In contrast to these approaches, we *explain how* proxy variables enhance OOD generalization.

**All variable models vs causal models.** Recently, Nastl & Hardt (2024) introduced a benchmark study where covariates are categorized into four groups: causal (conservatively chosen), arguably causal, anti-causal, and other spurious covariates. They show that across 16 benchmark datasets, models using all covariates Pareto-dominate those using only causal or arguably causal subsets on both ID and OOD data. However, there is limited theoretical work explaining these results. We present scenarios and arguments to explain their experimental findings. For linear causal models, anchor regression (Rothenhäusler et al., 2021) introduces a framework that balances between two estimation paradigms: models that include all observed covariates and models that focus solely on causal covariates. We aim to explain the impact of adding more covariates that are not necessarily causal under a hidden confounding shift. Eastwood et al. (2023) show that unstable covariates can boost performance when they carry information about the label, provided they are conditionally independent of the stable covariates given the label. They propose to adjust the distribution shift by looking at the test domain without labels. However, when applied to a medical real-world dataset not constructed for this particular problem (Bandi et al., 2019), ERM still remains competitive with their method, in line with the findings of Nastl & Hardt (2024). This reflects the broader insight that, under well-specified covariate shifts, maximum likelihood estimation (MLE) achieves minimax optimality for OOD generalization (Ge et al., 2024). Yet, real-world settings are rarely well-specified due to hidden confounding shift, which is the main focus of this work.

## 3 MANIFESTATIONS OF HIDDEN CONFOUNDING SHIFT

We now provide background on hidden confounding shift and motivate the need to address it.

**Types of distribution shifts.** For covariates $\mathbf{X}$ and target $Y$, one may observe several distribution shifts between two environments $e$ and $e'$ as shown in Table 1. These distribution shifts usually result from a shift in the distribution $\mathbb{P}(U)$ of an unobserved covariate $U$ that causes either $\mathbf{X}$ or $Y$ or both (see Figure 1). The shifts in $\mathbb{P}(U)$ lead to shifts in observed distributions involving only $\mathbf{X}$ and $Y$. For instance, we can write: $\mathbb{P}(\mathbf{X}) = \sum_U \mathbb{P}(U)\mathbb{P}(\mathbf{X} \mid U)$. Thus, when

Table 1: A summary of different distribution shifts.

| **Type of Shift** | **Mathematical Expression** |
|---|---|
| Label | $\mathbb{P}^e(Y) \neq \mathbb{P}^{e'}(Y)$ |
| Covariate | $\mathbb{P}^e(\mathbf{X}) \neq \mathbb{P}^{e'}(\mathbf{X})$ |
| Conditional Covariate | $\mathbb{P}^e(\mathbf{X} \mid Y) \neq \mathbb{P}^{e'}(\mathbf{X} \mid Y)$ |
| Concept | $\mathbb{P}^e(Y \mid \mathbf{X}) \neq \mathbb{P}^{e'}(Y \mid \mathbf{X})$ |
| Dataset | $\mathbb{P}^e(\mathbf{X}, Y) \neq \mathbb{P}^{e'}(\mathbf{X}, Y)$ |

$\mathbb{P}^e(U) \neq \mathbb{P}^{e'}(U)$ and $U \to \mathbf{X}$, we observe $\mathbb{P}^e(\mathbf{X}) \neq \mathbb{P}^{e'}(\mathbf{X})$. Similar arguments can be made about the distribution shifts of $\mathbb{P}(Y)$, $\mathbb{P}(\mathbf{X} \mid Y)$, $\mathbb{P}(Y \mid \mathbf{X})$, $\mathbb{P}(\mathbf{X}, Y)$.

Existing methods for OOD generalization assume certain causal relationships among $U, \mathbf{X}, Y$ that guarantee specific invariances. For instance, when $U \to Y \to \mathbf{X}$ (Figure 1 (a)), since $\mathbb{P}(\mathbf{X}, Y) = \mathbb{P}(Y)\mathbb{P}(\mathbf{X} \mid Y)$, a shift $\mathbb{P}^e(\mathbf{X}, Y) \neq \mathbb{P}^{e'}(\mathbf{X}, Y)$ is observed due to label shift, i.e., $\mathbb{P}^e(Y) \neq \mathbb{P}^{e'}(Y)$, but conditional covariate distribution stays

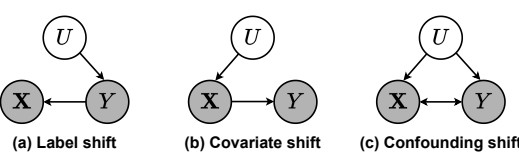

Figure 1: Causal graphs underlying distribution shifts.

invariant i.e., $\mathbb{P}^e(\mathbf{X} \mid Y) = \mathbb{P}^{e'}(\mathbf{X} \mid Y)$ because $\mathbb{P}(\mathbf{X}|Y, U) = \mathbb{P}(\mathbf{X}|Y)$ (Wu et al., 2021; Tachet des Combes et al., 2020; Garg et al., 2020; Alexandari et al., 2020). Similarly, when $U \to \mathbf{X} \to Y$ (Figure 1 (b)), a shift $\mathbb{P}^e(\mathbf{X}, Y) \neq \mathbb{P}^{e'}(\mathbf{X}, Y)$ is observed due to covariate shift, i.e., $\mathbb{P}^e(\mathbf{X}) \neq \mathbb{P}^{e'}(\mathbf{X})$, but the conditional distribution $\mathbb{P}(Y \mid \mathbf{X})$ stays invariant i.e., $\mathbb{P}^e(Y \mid \mathbf{X}) = \mathbb{P}^{e'}(Y \mid \mathbf{X})$ (Schneider et al., 2020; Sugiyama & Kawanabe, 2012; Gretton et al., 2009). These invariances may not hold in many scenarios because $U$ usually causes both $\mathbf{X}$ and $Y$ (Figure 1 (c)) (Liu et al., 2023; Alabdulmohsin et al., 2023; Landeiro & Culotta, 2018; Tsai et al., 2024; Prashant et al., 2025; Reddy et al., 2022; Reddy & N Balasubramanian, 2024). In this case, when $\mathbb{P}(U)$ shifts between environments, label shift, covariate shift, conditional covariate shift, and concept shift can all occur simultaneously. Nevertheless, confounding shift induces an invariance: $\mathbb{P}^e(\mathbf{X}, Y \mid U) = \mathbb{P}^{e'}(\mathbf{X}, Y \mid$

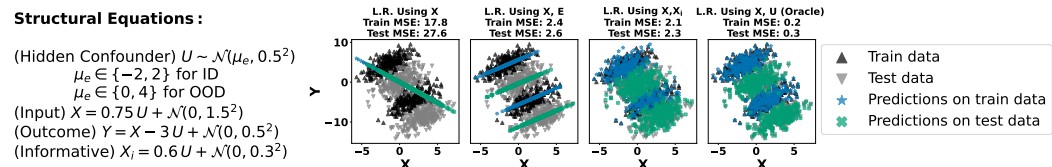

Figure 2: We evaluate four linear regression (L.R.) models in an OOD setting characterized by hidden confounding shifts and minimal environment overlap (i.e., distant $\mu_e$). (i) A model trained solely on $X$ learns an incorrect relationship with $Y$, illustrating Simpson's paradox. (ii) Using environment-specific summary statistics of $X$, denoted as $E$, recovers the correct relationship but remains limited in representation power. (iii) Using an informative covariate $X_i$ for $U$ improves OOD generalization. (iv) The oracle model is trained on $X$ and $U$.

$U$) referred to as *stable confounding shift* (Tsai et al., 2024; Alabdulmohsin et al., 2023). Since $U$ is unobserved, this invariance is absent from the observed data.

**Information-theoretic measures:** Following Federici et al. (2021), we use mutual information to quantify distribution shifts. That is, we use $I(\mathbf{X}; E)$, $I(Y; E)$, $I(\mathbf{X}; E|Y)$, $I(Y; E|\mathbf{X})$, $I(\mathbf{X}, Y; E)$ to measure the shifts in $\mathbb{P}(\mathbf{X})$, $\mathbb{P}(Y)$, $\mathbb{P}(\mathbf{X} \mid Y)$, $\mathbb{P}(Y \mid \mathbf{X})$, $\mathbb{P}(\mathbf{X}, Y)$ respectively across the environments $E$. As discussed earlier, existing methods for OOD generalization often assume either $I(\mathbf{X}; E \mid Y) = 0$ through the label shift assumption or $I(Y; E \mid \mathbf{X}) = 0$ through the covariate shift assumption. However both of them may be nonzero under hidden confounding shifts.

**Challenges with hidden confounding shift:** To illustrate the challenges posed by hidden confounding shift on OOD generalization, we present the experiment shown in Figure 2. The outcome $Y$ depends on $X$ and a hidden confounder $U$. Environment-specific shifts in the mean of $U$, denoted by $\mu_e$, induce distinct environments. A linear regression model trained solely on observed covariate $X$ infers an incorrect relationship between $X$ and $Y$, as shown in the first subplot. This exemplifies the Simpson's paradox (Simpson, 2018), where $Y$ increases with $X$ within each environment, yet the model captures the opposite trend. We observe similar behavior from the models designed for OOD generalization, such as IRM (Arjovsky et al., 2019), VREX (Krueger et al., 2021), and Group DRO (Sagawa et al., 2019). In such scenarios, we theoretically show in § 4 that the optimal strategy for generalization involves learning environment-specific relationships, which enables recovery of the correct relationship between $X$ and $Y$. This is illustrated in the second subplot, where environment-specific summary statistics of $X$, such as mean, standard deviation, and quantiles, help uncover the true $X$–$Y$ relationship. This mirrors the *backdoor adjustment* criteria in causal effect estimation (Pearl, 2009), where environment-specific information acts as a proxy for adjusting for confounders. However, as shown in the second subplot, environment-specific information encoded in observed covariates alone may be insufficient due to the limited representational capacity. However, as we show in § 4.3, additional informative covariates serving as proxies for unobserved confounders can further improve generalization performance, as shown in the third subplot. For comparison, we include an oracle model trained on $X$ and $U$, which achieves the best-possible mean squared error.

## 4 IMPACT OF HIDDEN CONFOUNDING SHIFTS ON OOD GENERALIZATION

In this section, we theoretically explain how hidden confounding shifts impact various aspects of OOD generalization. Let $\hat{Y} = (f \circ \phi)(\mathbf{X})$ be the predicted label for an input $\mathbf{X}$, where $f$ is a classifier and $\phi$ is a feature extractor or transformation function. $H(X) = -\mathbb{E}_X[\log(\mathbb{P}(X))]$ denotes entropy and $I(X; Y) = \mathbb{E}_{X,Y}[\log \frac{\mathbb{P}(X,Y)}{\mathbb{P}(X)\mathbb{P}(Y)}]$ denotes mutual information. The risk of the predictor $(f \circ \phi)$ in an environment $e$ is defined as: $\mathcal{R}^e(f \circ \phi) = \mathbb{E}_{(\mathbf{X},Y) \sim \mathbb{P}^e_{\mathbf{X},Y}}[\ell((f \circ \phi)(\mathbf{X}), Y)]$ where $\ell(\cdot, \cdot)$ is a loss function. The goal in OOD generalization is to learn a predictor $(f \circ \phi)$ that performs well on both ID and OOD data. To this end, various objectives have been considered in the literature. Robust optimization-based methods (Ben-Tal & Nemirovski, 2002; Sinha et al., 2017; Sagawa et al., 2019) aim to minimize the worst risk across all training environments: $\mathcal{R}^{\text{Rob}}(f \circ \phi) = \max_{e \in \mathcal{E}_{tr}} \mathcal{R}^e(f \circ \phi)$ where $\mathcal{E}_{tr}$ denotes the set of training environments. Invariant risk minimization (IRM) (Arjovsky et al., 2019) aims to minimize $\sum_{e \in \mathcal{E}_{tr}} \mathcal{R}^e(f \circ \phi)$ with the constraint that $f$ is a simultaneously optimal classifier across all environments. Empirical risk minimization (ERM) based methods simply

pool data from all training environments and minimize the empirical risk on the pooled data (Arjovsky et al., 2019; Krueger et al., 2021). That is, ERM minimizes $\mathcal{R}^{\text{ERM}}(f \circ \phi) = \sum_{e \in \mathcal{E}_{tr}} \mathcal{R}^e(f \circ \phi)$.

To understand how hidden confounding shift impacts traditional objective functions, we explore different aspects of maximizing predictive information: $I(Y; \hat{Y})$ where $\hat{Y} = (f \circ \phi)(\mathbf{X})$. $I(Y; \hat{Y})$ quantifies how informative the prediction $\hat{Y}$ is about the true label $Y$, making it a natural and meaningful objective for many tasks. We start by defining two key properties of $\phi(\mathbf{X})$—*informativeness* and *invariance*—both of which are crucial for OOD generalization (Ye et al., 2021).

**Definition 4.1** (Informativeness and Conditional Informativeness). *The informativeness of features $\phi(\mathbf{X})$ for predicting $Y$ is defined as $INF(\phi(\mathbf{X}), Y) = I(\phi(\mathbf{X}); Y)$. The conditional informativeness of features $\phi(\mathbf{X})$ for predicting $Y$ conditioned on environment variable $E$ is defined as $CINF(\phi(\mathbf{X}), Y, E) = I(\phi(\mathbf{X}); Y|E)$.*

Conditional informativeness measures the information $\phi(\mathbf{X})$ provides about $Y$ within each environment. Minimizing CINF implies information loss, and maximizing CINF can be undesirable in applications such as algorithmic fairness, as $\phi(\mathbf{X})$ may exploit sensitive or biased information within environments, leading to unfair predictions. While such biased representations can be avoided by minimizing $I(\phi(\mathbf{X}); E)$, it may reduce predictive performance, as we show in § 4.2. Considering the setting where no sensitive information is associated with $E$, we adopt the perspective that maximizing CINF can improve generalization performance, a view we follow and substantiate in § 4.1.

**Definition 4.2** (Variation and Invariance). *For a given label $Y$, the variation in the features $\phi(\mathbf{X})$ across environments $E$ is defined as $VAR(\phi(\mathbf{X}), Y, E) = I(\phi(\mathbf{X}); E|Y)$. For a given label $Y$, the invariance of the features $\phi(\mathbf{X})$ across environments $E$ is defined as the negative of the variation i.e., $INV(\phi(\mathbf{X}), Y, E) = -VAR(\phi(\mathbf{X}), Y, E) = -I(\phi(\mathbf{X}); E|Y)$*

Invariance quantifies how consistent the representation $\phi(\mathbf{X})$ is across different environments for a given label $Y$. Minimizing invariance can lead to overfitting by preserving environment-specific information in $\phi(\mathbf{X})$, whereas maximizing invariance helps eliminate environment-specific dependencies, promoting invariant learning. Extending the information-theoretic measures from § 3, we use $I(\phi(\mathbf{X}); E)$ and $I(Y; E \mid \phi(\mathbf{X}))$ to quantify *feature shift* and *concept shift* respectively. Depending on the context, we use the term *concept shift* to denote either $I(Y; E \mid \mathbf{X})$ or $I(Y; E \mid \phi(\mathbf{X}))$.

## 4.1 A GENERAL DECOMPOSITION OF PREDICTIVE INFORMATION

We begin with a few causal-graph preliminaries. A causal graph $\mathcal{G}$ consists of nodes representing random variables, and directed edges indicating direct causal influences between nodes. A *path* between two nodes $X_i$ and $X_j$ is a sequence of unique nodes connected by edges. A *directed path* from $X_i$ to $X_j$ with $i < j$ is one where all arrows point toward $X_j$, i.e., $X_i \to X_{i+1} \to \cdots \to X_{j-1} \to X_j$. In such a directed path, $X_i$ is called the *parent* of $X_{i+1}$, $X_{i+1}$ is the *child* of $X_i$, $X_i$ an *ancestor* of $X_j$, and $X_j$ is a *descendant* of $X_i$. Paths decompose into three fundamental structures: a *chain* $X_i \to X_j \to X_k$, a *fork* $X_i \leftarrow X_j \to X_k$, and a *collider* $X_i \to X_j \leftarrow X_k$. In both chains and forks, $X_i$ and $X_k$ are marginally dependent yet become conditionally independent upon conditioning on the intermediate node $X_j$, i.e. $X_i \not\perp\!\!\!\perp X_k$ and $X_i \perp\!\!\!\perp X_k \mid X_j$. In a collider, $X_i$ and $X_k$ are marginally independent but become conditionally dependent when conditioning on $X_j$ or any of its descendants, i.e. $X_i \perp\!\!\!\perp X_k$ and $X_i \not\perp\!\!\!\perp X_k \mid X_j$. A path between $X_i$ and $X_k$ is said to be *blocked* by a conditioning set $\mathcal{S}$ if and only if either (i) the path contains a chain or fork whose middle node lies in $\mathcal{S}$, or (ii) it contains a collider such that neither the collider nor any of its descendants belongs to $\mathcal{S}$. If all paths from $X_i$ to $X_k$ are blocked by $\mathcal{S}$, then $X_i \perp\!\!\!\perp X_k \mid \mathcal{S}$. A path is open if it is not blocked.

Now consider the predictive information $I(Y; \hat{Y})$, where the predictions $\hat{Y} = (f \circ \phi)(\mathbf{X})$ are based on a learned representations $\phi(\mathbf{X})$. We model the underlying causal relationships among $\mathbf{X}, Y, U, \phi(\mathbf{X}), \hat{Y}$ and $E$ as shown in Figure 3. Here, $U$ is a hidden confounding variable. Given either $\mathbf{X}$ or $\phi(\mathbf{X})$, $\hat{Y}$ is redundant for reasoning about $Y$. That is, $\hat{Y} \perp\!\!\!\perp Y \mid \mathbf{X}$ and $\hat{Y} \perp\!\!\!\perp Y \mid \phi(\mathbf{X})$. $E$ denotes an environment variable that captures shifts in $\mathbb{P}(U)$. That is, between

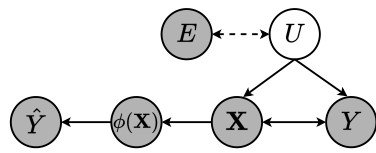

Figure 3: Bi-directed arrow between $\mathbf{X}$ and $Y$ indicate that some covariates cause $Y$, and some are caused by $Y$.

any two environments, there is a shift in $\mathbb{P}(U)$. *We first present a more general decomposition of predictive information without explicit consideration of how $U$ influences $\mathbf{X}, Y$. A formal treatment of $U$'s influence on $\mathbf{X}$ and $Y$ is presented in § 4.2.*

**Theorem 4.1.** *For a covariate vector $\mathbf{X}$, label $Y$, with causal structure $\mathbf{X} \leftrightarrow Y$, i.e., some covariates cause $Y$ and some covariates are caused by $Y$, environment variable $E$, a feature extractor $\phi$, and prediction $\hat{Y}$, the predictive information $I(Y; \hat{Y})$ is decomposed as follows:*

$$I(Y;\hat{Y}) = \overbrace{I(\phi(\mathbf{X});Y|E)}^{\text{Cond. informativ.}} - \frac{\overbrace{I(\phi(\mathbf{X});E|Y)}^{\text{Variation}}}{2} + \frac{\overbrace{I(Y;E)}^{\text{Label shift}}}{2} + \frac{\overbrace{I(\phi(\mathbf{X});E)}^{\text{Feature shift}}}{2} - \frac{\overbrace{I(Y;E|\phi(\mathbf{X}))}^{\text{Concept shift}}}{2} - \overbrace{I(\phi(\mathbf{X});Y|\hat{Y})}^{\text{Residual}} \tag{1}$$

*where $I(\phi(\mathbf{X}); Y|\hat{Y})$ is the residual information in $\phi(\mathbf{X})$ for inferring $Y$ that is not captured by the prediction $\hat{Y}$. The decomposition above also holds when $\phi(\mathbf{X})$ is replaced with $\mathbf{X}$.*

Proofs are presented in Appendix § A. Existing methods can be viewed as methods that explicitly minimize or maximize certain terms in Equation 1. For instance, IRM (Arjovsky et al., 2019) enforces that a fixed classifier $f$ remains the same across different environments. This is directly related to maximizing $-I(\phi(\mathbf{X}); E|Y)$ because if $\phi(\mathbf{X})$ were to carry extra environment-specific information conditional on $Y$, the fixed classifier would no longer be optimal in all environments. DANN (Ganin et al., 2016) and the independence criterion for fair classification (Federici et al., 2021) aims to minimize environment-specific information in $\phi(\mathbf{X})$ by minimizing $I(\phi(\mathbf{X}); E)$. In the context of domain adaptation, certain properties of features are learned invariant to domains (Sun & Saenko, 2016b; Xu et al., 2020). While minimizing $I(\phi(\mathbf{X}); E)$ may be good for fair classification, it may degrade predictive performance (Johansson et al., 2019; Federici et al., 2021). This can be seen from Equation 1, where minimizing $I(\phi(\mathbf{X}); E)$ can reduce $I(Y; \hat{Y})$. In CDAN (Long et al., 2018), the objective is to obtain $\mathbb{P}^e(\phi(\mathbf{X}), Y) = \mathbb{P}^{e'}(\phi(\mathbf{X}), Y)$. By enforcing this equality, $E$ becomes independent of the pair $(\phi(\mathbf{X}), Y)$. Consequently, $\mathbb{P}(Y \mid \phi(\mathbf{X}), E) = \mathbb{P}(Y \mid \phi(\mathbf{X})) \implies I(Y; E \mid \phi(\mathbf{X})) = 0$. Thus, although CDAN does not explicitly include $I(Y; E \mid \phi(\mathbf{X}))$ in its loss, its joint-distribution alignment objective effectively drives $I(Y; E \mid \phi(\mathbf{X}))$ to zero. While $I(Y; E)$ is a constant, there exist methods that are robust to label shift (Sagawa et al., 2019).

Recall that Equation 1 is derived without explicitly considering how $U$ influences $\mathbf{X}, Y$. Knowing how $U$ influences $\mathbf{X}$ and $Y$ can further guide better understanding of the terms in Equation 1. That is, if $U \to Y, U \not\to \mathbf{X}, Y \to \mathbf{X}$, then $I(\mathbf{X}; E \mid Y) = 0$, suggesting that minimizing $I(\phi(\mathbf{X}); E \mid Y)$ is a principled objective. Similarly, if $U \not\to Y, U \to \mathbf{X}, \mathbf{X} \to Y$, then $I(Y; E \mid \mathbf{X}) = 0$, thereby motivating the minimization of $I(Y; E \mid \phi(\mathbf{X}))$. However, if $U \to Y$ and $U \to \mathbf{X}$, the interaction between the terms in Equation 1 becomes non-trivial, and it remains unclear what constitutes an ideal strategy for addressing them. We answer this question in the next section.

## 4.2 PREDICTIVE INFORMATION UNDER HIDDEN CONFOUNDING SHIFT

To understand how hidden confounding variable $U$ that cause both $\mathbf{X}$ and $Y$, impact the predictive information, we consider *two special cases* of the relationship between $\mathbf{X}$ and $Y$: (i) $\mathbf{X} \to Y$ and (ii) $Y \to \mathbf{X}$. When $\mathbf{X} \to Y$, we obtain the inequalities in 2 because conditioning on $Y$ opens the path $\phi(\mathbf{X}) \leftarrow \mathbf{X} \to Y \leftarrow U \leftarrow E$ from $\phi(\mathbf{X})$ to $E$ at the collider node $Y$ which results in additional information flow from $\phi(\mathbf{X})$ to $E$ via $Y$. In contrast, conditioning on $\phi(\mathbf{X})$ partially blocks the information flow from $Y$ to $E$ at the node $\mathbf{X}$ as long as $\phi(\mathbf{X})$ encodes some information about $\mathbf{X}$.

$$(i)\ \overbrace{I(\phi(\mathbf{X});E|Y)}^{\text{Variation}} \geq \overbrace{I(\phi(\mathbf{X});E)}^{\text{Feature shift}} \qquad\qquad (ii)\ \overbrace{I(Y;E)}^{\text{Label shift}} \geq \overbrace{I(Y;E|\phi(\mathbf{X}))}^{\text{Concept shift}} \tag{2}$$

Similarly, when $Y \to \mathbf{X}$, we obtain the inequalities in 3 because conditioning on $Y$ blocks the information flow from $\phi(\mathbf{X})$ to $E$ at the node $Y$ and conditioning on $\phi(\mathbf{X})$ opens the path $Y \to \mathbf{X} \leftarrow U \leftarrow E$ from $Y$ to $E$ at the node $\mathbf{X}$ because $\phi(\mathbf{X})$ is the child of $\mathbf{X}$ and conditioning on the child of a collider opens a path via that collider (Pearl, 2009) (recall the preliminaries in § 4.1).

$$(i)\ \overbrace{I(\phi(\mathbf{X});E)}^{\text{Feature shift}} \geq \overbrace{I(\phi(\mathbf{X});E|Y)}^{\text{Variation}} \qquad\qquad (ii)\ \overbrace{I(Y;E|\phi(\mathbf{X}))}^{\text{Concept shift}} \geq \overbrace{I(Y;E)}^{\text{Label shift}} \tag{3}$$

Using the inequalities in 2, 3, we refine the predictive information decomposition as follows.

**Theorem 4.2.** *For a covariate vector* $\mathbf{X}$*, label* $Y$*, an environment variable* $E$*, a feature extractor* $\phi$*, the prediction* $\hat{Y}$*, and an unobserved confounding variable* $U$ *that cause both* $\mathbf{X}$ *and* $Y$*, and if either (i)* $\mathbf{X} \to Y$ *or (ii)* $Y \to \mathbf{X}$*, the predictive information* $I(Y; \hat{Y})$ *can be decomposed as follows:*

$$I(Y; \hat{Y}) = \overbrace{I(\phi(\mathbf{X}); Y | E)}^{\textit{Cond. informativeness}} - \overbrace{I(\phi(\mathbf{X}); Y | \hat{Y})}^{\textit{Residual}} \tag{4}$$

In Theorem 4.2, we consider the two cases: $\mathbf{X} \to Y$ and $Y \to \mathbf{X}$ separately. However, the causal relationship $\mathbf{X} \to Y$ is more prevalent in real-world tabular prediction tasks. For instance, 11 out of 16 datasets considered in recent benchmark studies (Nastl & Hardt, 2024; Liu et al., 2023; Gardner et al., 2023) follow the causal structure $\mathbf{X} \to Y$. In the remaining 5 out of 16 datasets, the number of covariates that are caused by $Y$ is much smaller than the number of covariates that cause $Y$. We present the key understanding from Theorem 4.2 below.

> Compared to the decomposition in 1, the decomposition in 4 (which is obtained for the special cases: (i) $U \to \mathbf{X}, U \to Y, \mathbf{X} \to Y$, (ii) $U \to \mathbf{X}, U \to Y, Y \to \mathbf{X}$) is free from the terms: *variation, label shift, feature shift, and concept shift*. That is, under hidden confounding shift, when either $\mathbf{X} \to Y$ or $Y \to \mathbf{X}$, the predictive information is equal to the difference: *conditional informativeness* $-$ *residual*.

**Remarks on conditional informativeness:** In a very recent work, Prashant et al. (2025) proposed a mixture of experts (MoE) model for OOD generalization under hidden confounding shift, where each expert corresponds to a specific hidden confounder assignment. That is, each expert focuses on maximizing the performance within the environment corresponding to a particular value of the hidden confounder. Thus, learning MoE models aligns with the goal of maximizing the conditional informativeness $I(\phi(\mathbf{X}); Y | E)$. Equation 4 provides theoretical justification for the use of MoE-type models. Beyond supporting such models, the Equation 4 also highlights the need for methods that operate under more general confounding shift settings. For instance, the method proposed by Prashant et al. (2025) assumes overlapping confounding support i.e., $\text{supp}(\mathbb{P}(U|\mathcal{E}_{te})) \subseteq \text{supp}(\mathbb{P}(U|\mathcal{E}_{tr}))$, and the proxy variables to be discrete-valued.

> The results in Figure 2 illustrate that, in a simple setting without confounding overlap and with a continuous-valued proxy, a linear regression model can successfully recover the correct causal relationship between observed covariates and the target across environments when provided with environment-specific information. We theoretically explain in § 4.3 why environment-specific informative covariates help improve generalization performance.

**Remarks on residual:** The residual term $I(\phi(\mathbf{X}); Y | \hat{Y})$ quantifies how much additional information $\phi(\mathbf{X})$ provides about the true label $Y$ beyond what is already contained in the prediction $\hat{Y}$. The residual term can be expressed as: $I(\phi(\mathbf{X}); Y | \hat{Y}) = H(Y | \hat{Y}) - H(Y | \phi(\mathbf{X}), \hat{Y})$. The conditional entropy $H(Y | \hat{Y})$ is related to the expected cross-entropy loss as: $\mathbb{E}_{Y, \hat{Y}}[\ell_{\text{CE}}(Y, \hat{Y})] = H(Y | \hat{Y}) + \textit{calibration error}$ (Bröcker, 2009; Berta et al., 2025). Here $H(Y | \hat{Y})$ is known as the *refinement error* that measures the model's ability to distinguish between classes.

### 4.3 Impact of additional informative covariates

In § 4.2, we present the desiderata for generalization under hidden confounding shifts. In practice, we sometimes have access to proxies for the hidden confounding variable (Alabdulmohsin et al., 2023; Tsai et al., 2024; Prashant et al., 2025), which can be leveraged to substitute for the hidden confounding variable. We now explain how adding such non-causal but informative covariates improves predictive information.

**Definition 4.3** (Informative Covariates). *A set of covariates* $\mathbf{X}_I$ *that are not causally related to* $Y$ *i.e., neither ancestors nor descendants of* $Y$ *are in* $\mathbf{X}_I$*, are said to be informative to* $Y$ *if* $\mathbf{X}_I$ *and* $Y$ *are not independent of each other given other causally related covariates* $\mathbf{X}$ *and* $E$ *i.e.,* $Y \not\perp\!\!\!\perp \mathbf{X}_I | \mathbf{X}, E$.

Since $U \to Y$, any covariate that is informative to $U$, is also informative to $Y$ and vice-versa. Causal graphs that show informative covariates $\mathbf{X}_I$ have the structure $U \leftrightarrow \mathbf{X}_I; Y \leftrightarrow \mathbf{X}_I$ in addition

Table 2: Quantifying distribution shifts. Mean±standard deviation is computed over 10 random subsets of 40,000 samples. Statistical significance against a mean of zero is assessed via one-sample t-tests ($\alpha = 0.05$), confirming all measures are significantly different from zero (p-value $\approx 0$).

| Dataset | Conditional Covariate Shift $I(\mathbf{X}; E\|Y)$ | Label shift $I(Y; E)$ | Covariate Shift $I(\mathbf{X}; E)$ | Concept shift $I(Y; E\|\mathbf{X})$ |
|---|---|---|---|---|
| Readmission | $0.107 \pm 0.002$ | $0.068 \pm 0.002$ | $0.097 \pm 0.002$ | $2.032 \pm 0.000$ |
| Food stamps | $0.126 \pm 0.004$ | $0.030 \pm 0.003$ | $0.108 \pm 0.001$ | $2.118 \pm 0.001$ |
| Income | $0.168 \pm 0.002$ | $0.075 \pm 0.003$ | $0.147 \pm 0.001$ | $2.059 \pm 0.002$ |
| Public coverage | $0.231 \pm 0.002$ | $0.412 \pm 0.006$ | $0.222 \pm 0.002$ | $1.945 \pm 0.001$ |
| Unemployment | $0.117 \pm 0.001$ | $0.019 \pm 0.002$ | $0.114 \pm 0.002$ | $2.010 \pm 0.003$ |
| Diabetes | $0.032 \pm 0.002$ | $0.048 \pm 0.001$ | $0.022 \pm 0.002$ | $2.132 \pm 0.001$ |
| Hypertension | $0.090 \pm 0.002$ | $0.183 \pm 0.003$ | $0.037 \pm 0.001$ | $1.883 \pm 0.004$ |
| ASSISTments | $0.293 \pm 0.002$ | $0.260 \pm 0.001$ | $0.306 \pm 0.002$ | $0.367 \pm 0.004$ |

to the causal relationships shown in Figure 3. From the predictive information decomposition in Equation 1, to maximize $I(Y; \hat{Y})$, it is required to minimize the concept shift $I(Y; E \mid \phi(\mathbf{X}))$. Liu et al. (2023) suggest based on their empirical analysis that collecting additional covariates $\mathbf{X}_I$ such that $\mathbb{P}(Y|\mathbf{X}, \mathbf{X}_I)$ is more stable across environments i.e., reducing concept shift, improves the OOD test accuracy. We theoretically show that utilizing more informative variables helps maximizing predictive information by maximizing or minimizing certain terms in Equation 1.

**Proposition 4.1.** *If $\phi_1(\cdot), \phi_2(\cdot)$ are invertible functions and if $\mathbf{X}_I$ are informative variables such that $Y \not\perp\!\!\!\perp \mathbf{X}_I \mid \mathbf{X}, E$, then we have the following inequalities:*

$$(i)\ I(\phi_2(\{\mathbf{X} \cup \mathbf{X}_I\}); Y|E) > I(\phi_1(\mathbf{X}); Y|E) \quad (ii)\ I(\phi_2(\{\mathbf{X} \cup \mathbf{X}_I\}); E) > I(\phi_1(\mathbf{X}); E)$$
$$(iii)\ I(Y; E|\phi_2(\{\mathbf{X} \cup \mathbf{X}_I\})) < I(Y; E|\phi_1(\mathbf{X})) \quad (iv)\ I(\phi_2(\{\mathbf{X} \cup \mathbf{X}_I\}); E \mid Y) > I(\phi_1(\mathbf{X}); E \mid Y) \tag{5}$$

Proposition 4.1 shows that adding informative covariates increases conditional informativeness (5.$i$) and feature shift (5.$ii$), while reducing concept shift (5.$iii$). However, adding informative covariates also amplifies variation (5.$iv$), which may necessitate dedicated strategies to control the variation. However, from our experimental results, we observe that the reduction of concept shift leads to significant improvement in the performance of models compared to minimizing variation.

## 5 EXPERIMENTAL RESULTS

We conduct experiments on both real-world and synthetic datasets to analyze: (i) the presence of hidden confounding shift in real-world data, (ii) how the components of the decomposition in 1 affect OOD generalization under hidden confounding shifts, and (iii) the role of informative covariates in improving generalization. We consider eight real-world tabular benchmark datasets: *Food stamps, Readmission, Income, Public coverage, Unemployment, Diabetes, Hypertension, and ASSISTments*. These datasets and corresponding domain splits are adopted from *TableShift* benchmark (Gardner et al., 2023). We use ID test and OOD test accuracies to measure the performance of models. We perform experiments on two ERM-based methods: XGBoost (XGB) (Chen & Guestrin, 2016) and multi-layer perceptron (MLP), two domain generalization methods: IRM (Arjovsky et al., 2019), VREX (Krueger et al., 2021), and one robust learning based method: Group DRO (GDRO) (Sagawa et al., 2019).

We use Non-Parametric Entropy Estimation toolbox (NPEET) for estimating mutual information (MI) via Kraskov-Stögbauer-Grassberger (KSG) estimator (Kraskov et al., 2004; Steeg & Galstyan, 2011; 2013). Following (Gardner et al., 2023), we evaluate mutual information using both in-distribution (ID) test data (from training domains) and out-of-distribution (OOD) test data (from test domains). Additional details of the experimental setup are provided in Appendix § B. The code and instructions to reproduce the results are provided in the supplementary material.

**Hidden confounding shift:** Recall that hidden confounding shifts induce observable shifts in the distributions $\mathbb{P}(\mathbf{X}), \mathbb{P}(\mathbf{X} \mid Y), \mathbb{P}(Y)$, and $\mathbb{P}(Y \mid \mathbf{X})$. Results in Table 2 show that label shift, covariate shift, conditional covariate shift, and concept shift are all present in real-world datasets, indicating the presence of hidden confounding shifts. For the qualitative understanding, we query GPT-4o (Achiam et al., 2023) to list potential hidden confounders for several benchmark datasets (Gardner et al., 2023;

Table 3: For all methods, sign consistency value is high for concept shift (CS). However, the sign consistency metric for conditional informativeness (CI) is crucial for generalization according to Theorem 4.2, and XGB excels at this. Res: residual, Var: variation, FS: feature shift, C: causal, AC: arguably causal, A: all.

| Method | Sign Consistency Metric (↑) | | | | | ID-Test Accuracy (↑) | | | OOD-Test Accuracy (↑) | | |
|--------|------|------|------|------|------|-------|-------|-------|-------|-------|-------|
| | CI | Var | FS | CS | Res | C | AC | A | C | AC | A |
| XGB | **0.92** | 0.56 | 0.35 | 0.79 | 0.15 | **78.91** | **81.96** | **82.31** | **64.35** | **72.80** | **72.90** |
| MLP | 0.65 | 0.60 | 0.21 | 0.85 | 0.42 | 77.56 | 78.86 | 80.16 | 62.03 | 67.92 | 66.93 |
| GDRO | 0.69 | 0.50 | 0.25 | 0.90 | 0.35 | 77.87 | 80.15 | 76.20 | 61.95 | 66.64 | 65.87 |
| IRM | 0.71 | 0.58 | 0.23 | 0.88 | 0.31 | 61.68 | 63.38 | 64.67 | 61.14 | 61.18 | 62.75 |
| VREX | 0.52 | 0.71 | 0.19 | 0.85 | 0.58 | 58.74 | 64.69 | 62.75 | 60.40 | 65.57 | 65.21 |

Nastl & Hardt, 2024). These results are presented in Appendix § E. For instance, in the *food stamps* dataset, unmeasured factors such as *economic policies* specific to each state may influence both *household* income and *food stamp recipiency*. *GPT-4o returned results are solely meant for semantic insight and are not implicitly or explicitly used in other experiments.*

**Conditional informativeness vs. accuracy:** From Theorem 4.2, under hidden confounding shift, maximizing the difference *conditional informativeness − residual* is essential for maximizing predictive information. We observe that the difference *conditional informativeness − residual* is positively correlated with ID test and OOD test accuracies, as measured using the Spearman rank correlation coefficient ($\rho$) between accuracy and the difference: conditional informativeness − residual. For the results on five methods and eight datasets, we obtain $\rho = 0.93$ with respect to ID test accuracy, and $\rho = 0.80$ with respect to OOD test accuracy. As shown in the dataset-wise results in Appendix C, we observe that sum *−variation / 2 + label shift / 2 + feature shift / 2 − concept shift / 2* is closer zero under due to hidden confounding shift (Theorem 4.2).

**Informative covariates vs. accuracy:** We now study how inclusion of informative covariates helps in achieving better OOD generalization. To this end, we adopt the covariate partitioning from (Nastl & Hardt, 2024), where all available covariates are grouped into three nested covariate subsets: *causal covariates ⊆ arguably causal covariates ⊆ all covariates* (see Appendix § B). We evaluate how the terms in the decomposition in 1 affect ID and OOD performance when going from one covariate subset to another covariate subset. To this end, we use what we call the *sign consistency metric*, which works as follows. When we move from one covariate set to a larger covariate set, for any term in 1 with a positive coefficient, such as conditional informativeness, we count how often its value rises when accuracy improves. For negatively weighted terms, such as concept shift, we count how often its value decreases when accuracy improves. The metric is the fraction of observations where a term's change aligns with its *beneficial* direction, thus capturing how reliably it contributes to better generalization. See Appendix § B for the formal definition. Table 3 shows that both conditional informativeness and concept shift exhibit high sign consistency with accuracy gains. While reducing concept shift is crucial, XGB further enhances conditional informativeness compared to other methods, yielding additional performance improvements. Dataset-specific results are presented in Appendix § C.

Note that in real-world datasets, it is often challenging to verify whether *all available covariates* include *every relevant or sufficiently informative* variable. To further test our hypothesis that informative covariates enhance generalization, we conduct experiments on synthetic data with a known causal structure: $U \to X, U \to Y, X \to Y$, and $U \to \mathbf{X}_I$, where $\mathbf{X}_I$ denotes informative covariates caused by the hidden confounder $U$. We observe that including $\mathbf{X}_I$ leads to improvements in conditional informativeness and feature shift, while reducing concept shift, as shown in Figure 4. The synthetic data generation process and additional results are provided in Appendix § D.

## 6 DISCUSSION, LIMITATIONS, AND FUTURE WORK

We decompose the predictive information between model outputs and true outcomes to explain the factors limiting OOD generalization under the hidden confounding shift. We explain why simple methods such as XGBoost work better than invariance-based OOD generalization methods. We also explain how the addition of non-causal informative covariates helps improve the generalization performance of any method. Our goal is to explain these phenomena but not to provide any solution

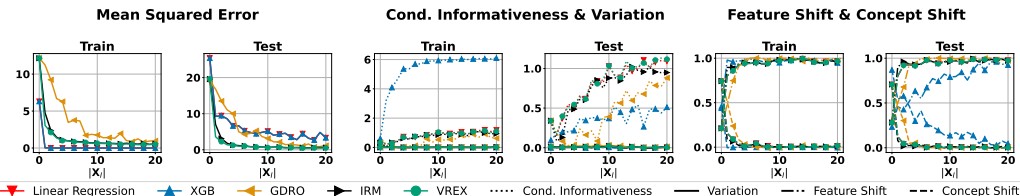

Figure 4: Adding more proxy variables $\mathbf{X}_I$ of $U$ that are informative to $Y$ helps in reducing MSE, increasing conditional informativeness and feature shift while reducing concept shift.

to the problems posed by the hidden confounding shift. By highlighting the inevitability of hidden confounding and the need to address it directly, our work lays a foundation for future work: (i) understanding the role of environments in maximizing conditional informativeness (ii) quantifying the cost–accuracy trade-off of acquiring non-causal informative covariates, (iii) handling entangled shifts without relying on untestable proxy assumptions, and (iv) modeling ambiguity from unobserved confounders to inspire new OOD-robust paradigms.

## ETHICS AND REPRODUCIBILITY STATEMENT

All authors have read and agree to adhere to the ICLR Code of Ethics. This work complies with all ethical guidelines outlined therein. Proofs of the theoretical results are presented in the appendix. The code and instructions to reproduce the results are available at `https://github.com/gautam0707/Confounding_is_to_Blame`.

## ACKNOWLEDGMENTS

The authors thank Anurag Singh and Rajeev Verma for proofreading the paper, which improved its readability. Moreover, the authors gratefully acknowledge the Gauss Centre for Supercomputing e.V. for funding this project by providing computing time on the GCS Supercomputer JUWELS at Jülich Supercomputing Centre (JSC). We also gratefully acknowledge funding from the European Research Council (ERC) under the Horizon Europe Framework Programme (HORIZON) for proposal number 101116395 SPARSE-ML.

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

APPENDIX

## A PROOFS OF THEORETICAL RESULTS

**Theorem 4.1.** *For a covariate vector $\mathbf{X}$, label $Y$, with causal structure $\mathbf{X} \leftrightarrow Y$, i.e., some covariates cause $Y$ and some covariates are caused by $Y$, environment variable $E$, a feature extractor $\phi$, and prediction $\hat{Y}$, the predictive information $I(Y; \hat{Y})$ is decomposed as follows:*

$$I(Y;\hat{Y}) = \overbrace{I(\phi(\mathbf{X});Y|E)}^{\text{Cond. informativ.}} - \overbrace{\frac{I(\phi(\mathbf{X});E|Y)}{2}}^{\text{Variation}} + \overbrace{\frac{I(Y;E)}{2}}^{\text{Label shift}} + \overbrace{\frac{I(\phi(\mathbf{X});E)}{2}}^{\text{Feature shift}} - \overbrace{\frac{I(Y;E|\phi(\mathbf{X}))}{2}}^{\text{Concept shift}} - \overbrace{I(\phi(\mathbf{X});Y|\hat{Y})}^{\text{Residual}} \tag{1}$$

*where $I(\phi(\mathbf{X});Y|\hat{Y})$ is the residual information in $\phi(\mathbf{X})$ for inferring $Y$ that is not captured by the prediction $\hat{Y}$. The decomposition above also holds when $\phi(\mathbf{X})$ is replaced with $\mathbf{X}$.*

*Proof.* From the causal graph shown in Figure 3 of the main paper, in the causal substructure: $\hat{Y} \leftarrow \phi(\mathbf{X}) \leftarrow \mathbf{X} \leftrightarrow Y$, $\hat{Y}$ provides no additional information in predicting $Y$ given either $\phi(\mathbf{X})$ or $\mathbf{X}$. This is because, the path $\hat{Y} \leftarrow \phi(\mathbf{X}) \leftarrow \mathbf{X} \leftrightarrow Y$, between $\hat{Y}$ and $Y$ is blocked by either $\{\phi(\mathbf{X})\}$ or $\{\mathbf{X}\}$. This implies $\hat{Y} \perp\!\!\!\perp Y|\phi(\mathbf{X})$ and $\hat{Y} \perp\!\!\!\perp Y|\mathbf{X}$. Now consider the expansion of $I(Y;\hat{Y})$.

$$\begin{aligned} I(Y;\hat{Y}) &= I(Y;\phi(\mathbf{X}),\hat{Y}) - I(\phi(\mathbf{X});Y|\hat{Y}) \qquad (\text{Introduce } \phi(\mathbf{X})) \\ &= I(\phi(\mathbf{X});Y) + \underbrace{I(Y;\hat{Y}|\phi(\mathbf{X}))}_{0} - I(\phi(\mathbf{X});Y|\hat{Y}) \qquad (\text{Expand and use } \hat{Y} \perp\!\!\!\perp Y \mid \phi(\mathbf{X})) \\ &= I(\phi(\mathbf{X}),E;Y) - I(Y;E|\phi(\mathbf{X})) - I(\phi(\mathbf{X});Y|\hat{Y}) \qquad (\text{Introduce } \mathbf{E}) \\ &= I(Y;E) + I(\phi(\mathbf{X});Y|E) - I(Y;E|\phi(\mathbf{X})) - I(\phi(\mathbf{X});Y|\hat{Y}) \qquad (\text{Expand}) \end{aligned} \tag{6}$$

Similarly,

$$\begin{aligned} I(Y;\hat{Y}) &= I(Y;\phi(\mathbf{X}),\hat{Y}) - I(\phi(\mathbf{X});Y|\hat{Y}) \qquad (\text{Introduce } \phi(\mathbf{X})) \\ &= I(\phi(\mathbf{X});Y) + \underbrace{I(Y;\hat{Y}|\phi(\mathbf{X}))}_{0} - I(\phi(\mathbf{X});Y|\hat{Y}) \qquad (\text{Expand and use } \hat{Y} \perp\!\!\!\perp Y \mid \phi(\mathbf{X})) \\ &= I(Y,E;\phi(\mathbf{X})) - I(\phi(\mathbf{X});E|Y) - I(\phi(\mathbf{X});Y|\hat{Y}) \qquad (\text{Introduce } \mathbf{E}) \\ &= I(\phi(\mathbf{X});E) + I(\phi(\mathbf{X});Y|E) - I(\phi(\mathbf{X});E|Y) - I(\phi(\mathbf{X});Y|\hat{Y}) \qquad (\text{Expand}) \end{aligned} \tag{7}$$

Summing Equations 6 and equation 7 and grouping the terms gives the desired expression:

$$\begin{aligned} I(Y;\hat{Y}) = I(\phi(\mathbf{X});Y|E) - \frac{I(\phi(\mathbf{X});E|Y)}{2} + \frac{I(Y;E)}{2} \\ + \frac{I(\phi(\mathbf{X});E)}{2} - \frac{I(Y;E|\phi(\mathbf{X}))}{2} - I(\phi(\mathbf{X});Y|\hat{Y}). \end{aligned}$$

To get the similar decomposition with $\mathbf{X}$ instead of $\phi(\mathbf{X})$, use $\hat{Y} \perp\!\!\!\perp Y \mid \mathbf{X}$ and introduce $\mathbf{X}$ into the expansion instead of $\phi(\mathbf{X})$. □

**Theorem 4.2.** *For a covariate vector $\mathbf{X}$, label $Y$, an environment variable $E$, a feature extractor $\phi$, the prediction $\hat{Y}$, and an unobserved confounding variable $U$ that cause both $\mathbf{X}$ and $Y$, and if either (i) $\mathbf{X} \to Y$ or (ii) $Y \to \mathbf{X}$, the predictive information $I(Y; \hat{Y})$ can be decomposed as follows:*

$$I(Y;\hat{Y}) = \overbrace{I(\phi(\mathbf{X});Y|E)}^{\text{Cond. informativeness}} - \overbrace{I(\phi(\mathbf{X});Y|\hat{Y})}^{\text{Residual}} \tag{4}$$

*Proof.* Following the causal graph shown in Figure 3 of the main paper, we prove the theorem in two separate cases: $\mathbf{X} \to Y$ and $Y \to \mathbf{X}$.

**Case 1: $\mathbf{X} \to Y$.** In this case, we have the following inequality because conditioning on $Y$ opens the path $\phi(\mathbf{X}) \leftarrow \mathbf{X} \to Y \leftarrow U \leftarrow E$ between $\phi(\mathbf{X})$ and $E$ at the collider node $Y$:

$$I(\phi(\mathbf{X}); E|Y) \geq I(\phi(\mathbf{X}); E). \tag{8}$$

Additionally, considering the causal substructure: $Y \leftarrow \mathbf{X} \leftarrow U \leftrightarrow E$, we have the following inequality because conditioning on $\phi(\mathbf{X})$ partially blocks the information flow from $Y$ to $E$ at the node $\mathbf{X}$ as long as $\phi(\mathbf{X})$ encodes some information about $\mathbf{X}$:

$$I(Y; E) \geq I(Y; E|\phi(\mathbf{X})). \tag{9}$$

Now consider the following expansion from Equation 7 in the proof of Theorem 4.1.

$$
\begin{aligned}
I(Y; \hat{Y}) &= I(\phi(\mathbf{X}); E) + I(\phi(\mathbf{X}); Y|E) - I(\phi(\mathbf{X}); E|Y) - I(\phi(\mathbf{X}); Y|\hat{Y}) && \text{(Equation 7)} \\
&\leq I(\phi(\mathbf{X}); Y|E) + I(\phi(\mathbf{X}); E|Y) - I(\phi(\mathbf{X}); E|Y) - I(\phi(\mathbf{X}); Y|\hat{Y}) && \text{(Using 8)} \\
&= I(\phi(\mathbf{X}); Y|E) - I(\phi(\mathbf{X}); Y|\hat{Y})
\end{aligned}
$$

Similarly, consider the following expansion from Equation 6 from the proof of Theorem 4.1:

$$
\begin{aligned}
I(Y; \hat{Y}) &= I(Y; E) + I(\phi(\mathbf{X}); Y|E) - I(Y; E|\phi(\mathbf{X})) - I(\phi(\mathbf{X}); Y|\hat{Y}) && \text{(Equation 6)} \\
&\geq I(Y; E) + I(\phi(\mathbf{X}); Y|E) - I(Y; E) - I(\phi(\mathbf{X}); Y|\hat{Y}) && \text{(Using 9)} \\
&= I(\phi(\mathbf{X}); Y|E) - I(\phi(\mathbf{X}); Y|\hat{Y})
\end{aligned}
$$

That is, we have $I(\phi(\mathbf{X}); Y|E) - I(\phi(\mathbf{X}); Y|\hat{Y}) \leq I(Y; \hat{Y}) \leq I(\phi(\mathbf{X}); Y|E) - I(\phi(\mathbf{X}); Y|\hat{Y})$.

**Case 2: $Y \to \mathbf{X}$.** In this scenario, we have the following inequality because conditioning on $\phi(\mathbf{X})$ opens the path $E \to U \to \mathbf{X} \leftarrow Y$ between $Y$ and $E$ at the collider node $\mathbf{X}$ because $\phi(\mathbf{X})$ is a child of $\mathbf{X}$ (Pearl, 2009).

$$I(Y; E|\phi(\mathbf{X})) \geq I(Y; E) \tag{10}$$

Additionally, we have the following inequality because conditioning on $Y$ blocks the path $\phi(\mathbf{X}) \leftarrow \mathbf{X} \leftarrow Y \leftarrow U \leftrightarrow E$ at $Y$ and hence less information flow between $\phi(\mathbf{X})$ and $E$.

$$I(\phi(\mathbf{X}); E) \geq I(\phi(\mathbf{X}); E|Y) \tag{11}$$

Now consider the following expansion from Equation 6 from the proof of Theorem 4.1.

$$
\begin{aligned}
I(Y; \hat{Y}) &= I(Y; E) + I(\phi(\mathbf{X}); Y|E) - I(Y; E|\phi(\mathbf{X})) - I(\phi(\mathbf{X}); Y|\hat{Y}) && \text{(Equation 6)} \\
&\leq I(\phi(\mathbf{X}); Y|E) + I(Y; E|\phi(\mathbf{X})) - I(Y; E|\phi(\mathbf{X})) - I(\phi(\mathbf{X}); Y|\hat{Y}) && \text{(Using 10)} \\
&= I(\phi(\mathbf{X}); Y|E) - I(\phi(\mathbf{X}); Y|\hat{Y})
\end{aligned}
$$

Similarly, consider the following expansion from Equation 7 from the proof of Theorem 4.1.

$$
\begin{aligned}
I(Y; \hat{Y}) &= I(\phi(\mathbf{X}); E) + I(\phi(\mathbf{X}); Y|E) - I(\phi(\mathbf{X}); E|Y) - I(\phi(\mathbf{X}); Y|\hat{Y}) && \text{(Equation 7)} \\
&\geq I(\phi(\mathbf{X}); Y|E) + I(\phi(\mathbf{X}); E) - I(\phi(\mathbf{X}); E) - I(\phi(\mathbf{X}); Y|\hat{Y}) && \text{(Using 11)} \\
&= I(\phi(\mathbf{X}); Y|E) - I(\phi(\mathbf{X}); Y|\hat{Y})
\end{aligned}
$$

That is, we have $I(\phi(\mathbf{X}); Y|E) - I(\phi(\mathbf{X}); Y|\hat{Y}) \leq I(Y; \hat{Y}) \leq I(\phi(\mathbf{X}); Y|E) - I(\phi(\mathbf{X}); Y|\hat{Y})$.

Since the upper bound and lower bounds are the same in both cases, we have the desired result $I(Y; \hat{Y}) = I(\phi(\mathbf{X}); Y|E) - I(\phi(\mathbf{X}); Y|\hat{Y})$. □

**Proposition 4.1.** *If $\phi_1(\cdot), \phi_2(\cdot)$ are invertible functions and if $\mathbf{X}_I$ are informative variables such that $Y \not\perp\!\!\!\perp \mathbf{X}_I \mid \mathbf{X}, E$, then we have the following inequalities:*

$$
\begin{aligned}
&(i)\; I(\phi_2(\{\mathbf{X} \cup \mathbf{X}_I\}); Y|E) > I(\phi_1(\mathbf{X}); Y|E) \quad (ii)\; I(\phi_2(\{\mathbf{X} \cup \mathbf{X}_I\}); E) > I(\phi_1(\mathbf{X}); E) \\
&(iii)\; I(Y; E|\phi_2(\{\mathbf{X} \cup \mathbf{X}_I\})) < I(Y; E|\phi_1(\mathbf{X})) \quad (iv)\; I(\phi_2(\{\mathbf{X} \cup \mathbf{X}_I\}); E \mid Y) > I(\phi_1(\mathbf{X}); E \mid Y)
\end{aligned} \tag{5}
$$

*Proof.* (i) By the chain rule of mutual information, we have: $I(\mathbf{X}, \mathbf{X}_I; Y|E) = I(\mathbf{X}; Y|E) + I(\mathbf{X}_I; Y|E, \mathbf{X})$. We have $I(\mathbf{X}_I; Y|E, \mathbf{X}) > 0$ because $Y \not\perp \mathbf{X}_I \mid \mathbf{X}, E$. Hence, we have $I(\mathbf{X}, \mathbf{X}_I; Y|E) > I(\mathbf{X}; Y|E)$. Since $\phi_1(\cdot), \phi_2(\cdot)$ are invertible functions and mutual information is invariant to invertible transformations, we have $I(\phi_2(\{\mathbf{X} \cup \mathbf{X}_I\}); Y|E) > I(\phi_1(\mathbf{X}); Y|E)$.

(ii) We consider the causal graph shown in Figure 3. From the chain rule of mutual information, we have: $I(\mathbf{X}, \mathbf{X}_I; E) = I(\mathbf{X}; E) + I(\mathbf{X}_I; E|\mathbf{X})$. Since $E \leftrightarrow U \rightarrow Y$, any covariate that is informative to $Y$ is also informative to $U$. Since $E \leftrightarrow U$, we have $I(\mathbf{X}_I; E|\mathbf{X}) > 0$, and hence $I(\mathbf{X}, \mathbf{X}_I; E) > I(\mathbf{X}; E)$. Since $\phi_1(\cdot), \phi_2(\cdot)$ are invertible functions and mutual information is invariant to invertible transformations, we have $I(\phi_2(\{\mathbf{X} \cup \mathbf{X}_I\}); E) > I(\phi_1(\mathbf{X}); E)$.

(iii) Consider $I(Y; E, \mathbf{X}_I|\mathbf{X})$. It can be expressed in two ways:

$$I(Y; E, \mathbf{X}_I \mid \mathbf{X}) = I(Y; E \mid \mathbf{X}) + I(Y; \mathbf{X}_I \mid E, \mathbf{X}) = I(Y; \mathbf{X}_I \mid \mathbf{X}) + I(Y; E \mid \mathbf{X}_I, \mathbf{X})$$

Equating and rearranging the terms gives the following.

$$I(Y; E \mid \mathbf{X}) - I(Y; E \mid \mathbf{X}_I, \mathbf{X}) = I(Y; \mathbf{X}_I \mid \mathbf{X}) - I(Y; \mathbf{X}_I \mid E, \mathbf{X})$$

Since $Y \not\perp \mathbf{X}_I \mid E, \mathbf{X}$, we have $I(Y; \mathbf{X}_I \mid \mathbf{X}) > I(Y; \mathbf{X}_I \mid E, \mathbf{X})$ because conditioning usually reduces mutual information unless the additional conditioning variable opens any collider paths. Here $E$ is neither a collider not a descendant of any collider. Thus, $I(Y; E \mid \mathbf{X}) > I(Y; E \mid \mathbf{X}_I, \mathbf{X})$. Since $\phi_1(\cdot), \phi_2(\cdot)$ are invertible functions and mutual information is invariant to invertible transformations, we have $I(Y; E \mid \phi_1(\mathbf{X})) > I(Y; E \mid \phi_2(\{\mathbf{X}_I \cup \mathbf{X}\}))$.

(iv) By the chain rule of mutual information, we have: $I(\mathbf{X}, \mathbf{X}_I; E \mid Y) = I(\mathbf{X}; E \mid Y) + I(\mathbf{X}_I; E|\mathbf{X}, Y)$. Since $E \leftrightarrow U \rightarrow Y$, any covariate that is informative to $Y$ is also informative to $U$. Since $E \leftrightarrow U$, we have $I(\mathbf{X}_I; E|\mathbf{X}, Y) > 0$, and hence $I(\mathbf{X}, \mathbf{X}_I; E \mid Y) > I(\mathbf{X}; E \mid Y)$. Since $\phi_1(\cdot), \phi_2(\cdot)$ are invertible functions and mutual information is invariant to invertible transformations, we have $I(\phi_2(\{\mathbf{X} \cup \mathbf{X}_I\}); E \mid Y) > I(\phi_1(\mathbf{X}); E \mid Y)$. □

# B EXPERIMENTAL SETUP

In this section, we detail our experimental setup. The impact of random seeds on the results is statistically insignificant in many settings (Gardner et al., 2023). For the results in the main paper, we report the mean and standard deviation across five random hyperparameter values (sampled from their respective domains in Table B1) for each dataset-method combination. Since we evaluate models on ID test and OOD test data, to evaluate mutual information terms, we use the nonparametric entropy estimation toolbox (Kraskov et al., 2004; Steeg & Galstyan, 2011; 2013). Following (Gardner et al., 2023), we evaluate mutual information using both in-distribution (ID) test data (from training domains) and out-of-distribution (OOD) test data (from test domains).

For consistency, we select 20,000 random samples from each ID and OOD test data, though some datasets contain fewer than 20,000 samples. Throughout our experiments, the sample size has a relatively insignificant effect on the evaluated mutual information terms. For instance, as shown in Figure B1, the terms in Equation 1 computed for the representations of the XGB model on the *income* dataset stayed relatively consistent as we vary sample size. Features $\phi(\mathbf{X})$ are extracted from the layer preceding the classification head; for XGBoost, we combine output margins and SHAP values from the model. All experiments were conducted on a single NVIDIA RTX A6000 GPU.

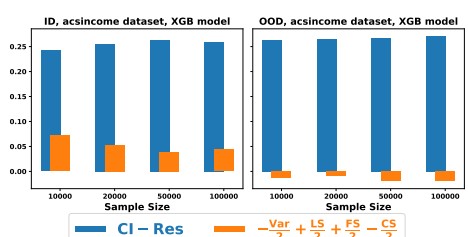

Figure B1: Evaluated terms of predictive information are relatively consistent across various sample sizes on the *income* dataset.

In our experiments studying the impact of informative covariates on predictive information, we follow the categorization introduced by Nastl & Hardt (2024), partitioning covariates into: *causal*, *arguably causal*, and *all covariates*. Causal covariates have clearly established influence on the target, with well-supported directionality and unlikely reverse causation. However, strict reliance on causal variables

Table B1: Hyperparameters and their possible values considered in this paper.

| Model | Hyperparameter | Search values |
|---|---|---|
| XGBoost | Learning rate | $[0.1, 0.2, 0.3]$ |
| | Maximum tree depth | $[4, 5, 6]$ |
| | Minimum child weight | $[0.1, 1, 10]$ |
| | Gamma (min. loss reduction) | $[0.0001, 0.001, 0.01]$ |
| | Subsample ratio | $[0.5, 0.6, 0.7]$ |
| | Column subsample per tree | $[0.5, 0.6, 0.7]$ |
| | L1 regularization ($\alpha$) | $[0.0001, 0.001, 0.01]$ |
| | L2 regularization ($\lambda$) | $[0.0001, 0.001, 0.01]$ |
| MLP | Number of layers | $[2, 3, 4]$ |
| | Hidden layer size | $[256, 512, 1024]$ |
| | Dropout rate | $[0.0, 0.1, 0.2]$ |
| | Learning rate | $[0.01, 0.02, 0.05]$ |
| | Weight decay (L2) | $[0.0001, 0.001, 0.01]$ |
| | Batch size | $[4096]$ |
| | Number of epochs | $[1, 2, 3]$ |
| GroupDRO | Number of layers | $[2, 3, 4]$ |
| | Hidden layer size | $[256, 512, 1024]$ |
| | Group-weights step size | $[0.01, 0.05, 0.1]$ |
| | Dropout rate | $[0.0, 0.1, 0.2]$ |
| | Learning rate | $[0.01, 0.02, 0.05]$ |
| | Weight decay (L2) | $[0.0001, 0.001, 0.01]$ |
| | Batch size | $[4096]$ |
| | Number of epochs | $[1, 2, 3]$ |
| IRM | Number of layers | $[2, 3, 4]$ |
| | Hidden layer size | $[256, 512, 1024]$ |
| | Dropout rate | $[0.0, 0.1, 0.2]$ |
| | IRM penalty weight ($\lambda$) | $[0.01, 0.05, 0.1]$ |
| | IRM penalty anneal iterations | $[1, 2, 3]$ |
| | Learning rate | $[0.01, 0.02, 0.05]$ |
| | Weight decay (L2) | $[0.0001, 0.001, 0.01]$ |
| | Batch size | $[4096]$ |
| | Number of epochs | $[1, 2, 3]$ |
| VREX | Number of layers | $[2, 3, 4]$ |
| | Hidden layer size | $[256, 512, 1024]$ |
| | VREX penalty anneal iterations | $[1, 2, 3]$ |
| | VREX penalty weight ($\lambda$) | $[0.1, 10, 100]$ |
| | Dropout rate | $[0.0, 0.1, 0.2]$ |
| | Learning rate | $[0.01, 0.02, 0.05]$ |
| | Weight decay (L2) | $[0.0001, 0.001, 0.01]$ |
| | Batch size | $[4096]$ |
| | Number of epochs | $[1, 2, 3]$ |

risks omitting relevant parents due to knowledge gaps. Arguably causal covariates have uncertain causal relationships, meeting at least one of: (1) being known causal, (2) having plausible but potentially bidirectional influence, or (3) likely (but unconfirmed) causal effect. These groups approximate true causal parents based on available knowledge, though relationships may be confounded. Some datasets additionally contain *anti-causal* covariates where the target likely affects them but not vice versa. The *all covariates* set includes all observed variables regardless of causal status. The sign consistency metric introduced in the main paper § 5 is defined as follows. For each $(m, \sigma_m) \in M$ where $M = \{(\text{conditional informativeness}, +1), (\text{variation}, -1), (\text{label shift}, +1), (\text{feature shift}, +1),$

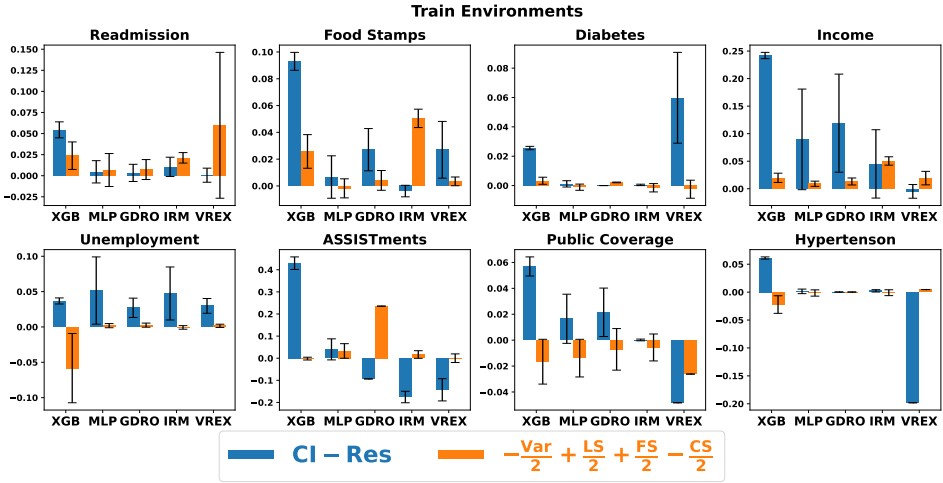

Figure C2: Dataset specific results on ID data.

(concept shift, $-1$), and (residual, $-1$)}, for a specific dataset, we compute:

$$C_m = \frac{1}{2|P|} \sum_{(s,t)\in P} \left[ \mathbb{I}\left( \sigma_m \cdot \left( \mu_t^{\text{tr}}(m) - \mu_s^{\text{tr}}(m) \right) > 0 \right) + \mathbb{I}\left( \sigma_m \cdot \left( \mu_t^{\text{te}}(m) - \mu_s^{\text{te}}(m) \right) > 0 \right) \right]$$

Where $P$ is the set of covariate set pairs $(s,t)$ with $t \supset s$. For example, $s$ can be causal covariates and $t$ can be arguably causal covariates. There is a factor of 2 in $2P$ that accounts for both train and test data. $\mu_s^{\text{tr/te}}(m)$ is the mean of measure $m$ for covariate set $s$ in training/testing data. $\mathbb{I}(\cdot)$ is the indicator function. $|P| = 3$ for covariate settings {*causal*, *arguably causal*, *all*}. To get final sign consistency metric value for each $m$, we average across datasets: $\bar{C}_m = \frac{1}{|D|} \sum_{d \in D} C_m^{(d)}$ where $|D|$ is the number of datasets.

## C  ADDITIONAL RESULTS ON REAL-WORLD DATASETS

In this section, we present additional results on real-world datasets. Figures C2, C3 show the comparison of the terms: *conditional informativeness − residual* and *−variation / 2 + label shift / 2 + feature shift / 2 − concept shift / 2* for each dataset. The key takeaway from these results is that, from Theorem 4.1, the sum of blue and orange terms is positively correlated with the overall model performance. At the same time, due to potential hidden confounding shift in real-world data, the contribution of the difference *conditional informativeness − residual* is higher towards the overall predictive information when compared to the contribution of *−variation / 2 + label shift / 2 + feature shift / 2 − concept shift / 2*. This serves as a motivation to maximize conditional informativeness for better OOD generalization under a hidden confounding shift.

Table C2: Comparison of ID and OOD test accuracies of models.

| Method | Readmission | | Food stamps | | Diabetes | |
|---|---|---|---|---|---|---|
| | ID | OOD | ID | OOD | ID | OOD |
| MLP | $65.78 \pm 0.14$ | $61.75 \pm 0..22$ | $84.74 \pm 0.03$ | $82.00 \pm 0.06$ | $87.66 \pm 0.02$ | $83.22 \pm 0.05$ |
| GDRO | $60.47 \pm 1.08$ | $57.60 \pm 0.42$ | $84.47 \pm 0.04$ | $81.41 \pm 0.09$ | $87.47 \pm 0.10$ | $82.90 \pm 0.10$ |
| IRM | $50.28 \pm 7.35$ | $50.85 \pm 1.74$ | $80.91 \pm 0.00$ | $78.01 \pm 0.00$ | $42.54 \pm 36.52$ | $43.48 \pm 31.92$ |

| Method | Income | | Unemployment | |
|---|---|---|---|---|
| MLP | $82.92 \pm 0.05$ | $81.46 \pm 0.01$ | $97.27 \pm 0.02$ | $96.04 \pm 0.05$ |
| GDRO | $82.70 \pm 0.05$ | $80.16 \pm 0.45$ | $97.28 \pm 0.01$ | $96.10 \pm 0.00$ |
| IRM | $67.91 \pm 0.00$ | $60.20 \pm 0.00$ | $96.61 \pm 0.00$ | $94.84 \pm 0.00$ |

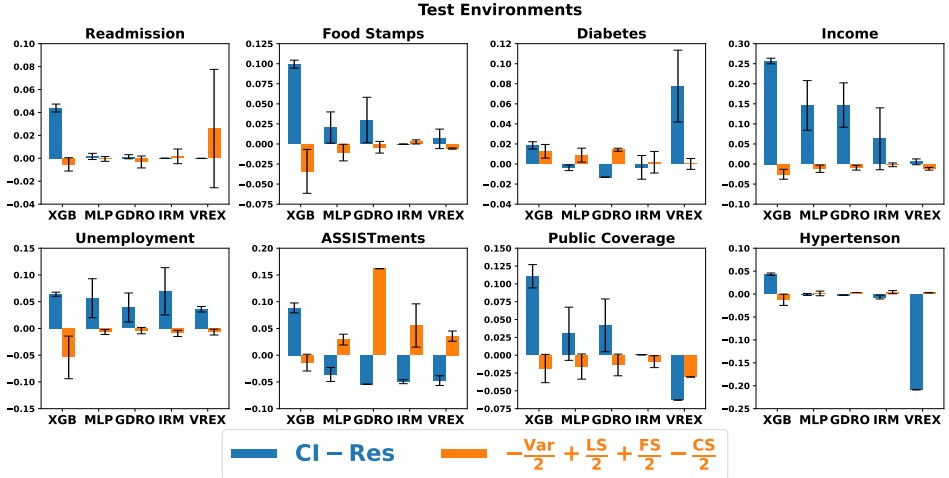

Figure C3: Dataset-specific results on OOD data.

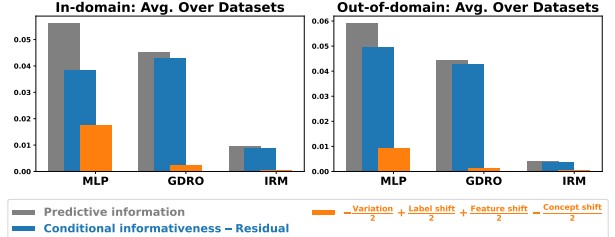

| Method | Avg. ID Test Acc. | Avg. OOD Test Acc. |
|--------|-------------------|--------------------|
| MLP    | 83.67             | 80.89              |
| GDRO   | 82.48             | 79.64              |
| IRM    | 67.65             | 65.48              |

Figure C4: The difference *conditional informativeness − residual* in the plots is positively correlated with the average ID and OOD test accuracy over five datasets shown in the table on the right. The contribution of the difference *conditional informativeness − residual* is higher towards the overall predictive information when compared to the contribution of −*variation / 2 + label shift / 2 + feature shift / 2 − concept shift / 2*.

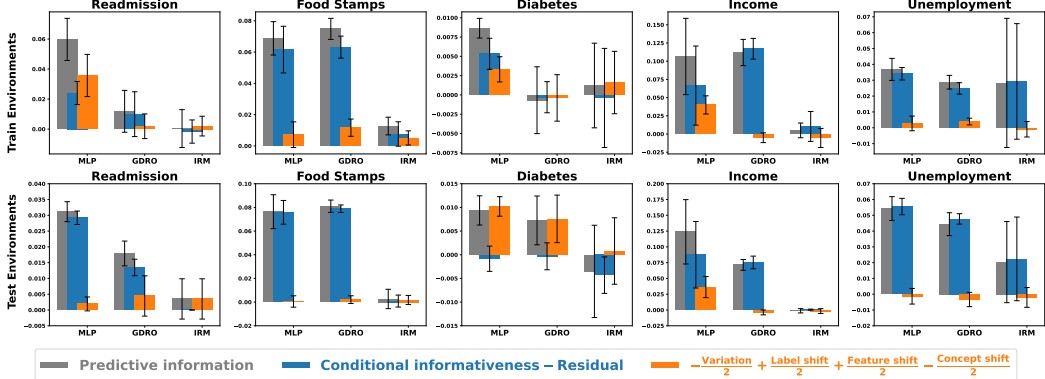

Figure C5: Dataset specific results. Comparison of predictive information among models.

Figures C6- C15 further show the comparison of various terms of the information decomposition in 1 for each method and dataset. From these results, we observe that the values plotted in the second column (variation) and third column (feature shift) are similar in magnitude, and based on Equation 1, their difference is close to zero, contributing very little to the overall predictive decomposition. Similarly, we observe that the values plotted in the fourth column (label shift) and fifth column (concept shift) are similar in magnitude, and based on Equation 1, their difference is close to zero, contributing very little to the overall predictive decomposition. In almost all settings, the difference *conditional informativenss - residual* is the majority contributor for the predictive information.

**Hyperparameter tuning:** We also perform hyperparameter tuning on model parameters. We run a grid search over hyperparameter values and select the best hyperparameter values based on ID validation accuracy (not OOD validation accuracy). We run 1500 experiments (5 datasets, 3 models, and 100 hyperparameter choices sampled from a grid of hyperparameter values). In this case, to get the mean and standard deviation results for accuracies and mutual information terms, we run each experiment for 5 random seeds, even if the effect of random seeds is insignificant in many cases. Results are shown in Figure C4. From Theorem 4.1, the sum of blue and orange terms is positively correlated with the overall model performance. At the same time, due to potential hidden confounding shift in real-world data, the contribution of the difference *conditional informativeness − residual* is higher towards the overall predictive information when compared to the contribution of *−variation / 2 + label shift / 2 + feature shift / 2 − concept shift / 2*. Dataset-specific results shown in Figure C5 and Table C2 also show similar trends.

**Potential limitations with predictive information decomposition analysis:** While predictive information $I(Y; \hat{Y})$ measures statistical dependence between predictions and labels, it does not guarantee accuracy. For instance, a binary classifier that systematically predicts the *wrong* label (e.g., $\hat{Y} = 1$ when $Y = 0$) achieves maximal $I(Y; \hat{Y}) = 1$ bit (for binary variables) due to perfect anti-correlation, yet yields 0% accuracy. This occurs because mutual information quantifies *reduction in uncertainty* rather than correctness. Other decomposition terms provide further insight: conditional informativeness $I(\phi(\mathbf{X}); Y | E)$ may mask harmful variation $I(\phi(\mathbf{X}); E | Y)$, while feature shift $I(\phi(\mathbf{X}); E)$ can inflate $I(Y; \hat{Y})$ through spurious correlations. Nevertheless, this decomposition remains valuable for analyzing reasonably performant models, as our experiments demonstrate - it reveals whether predictive information stems from generalizable patterns or environment-specific artifacts, enabling targeted improvements while maintaining interpretability.

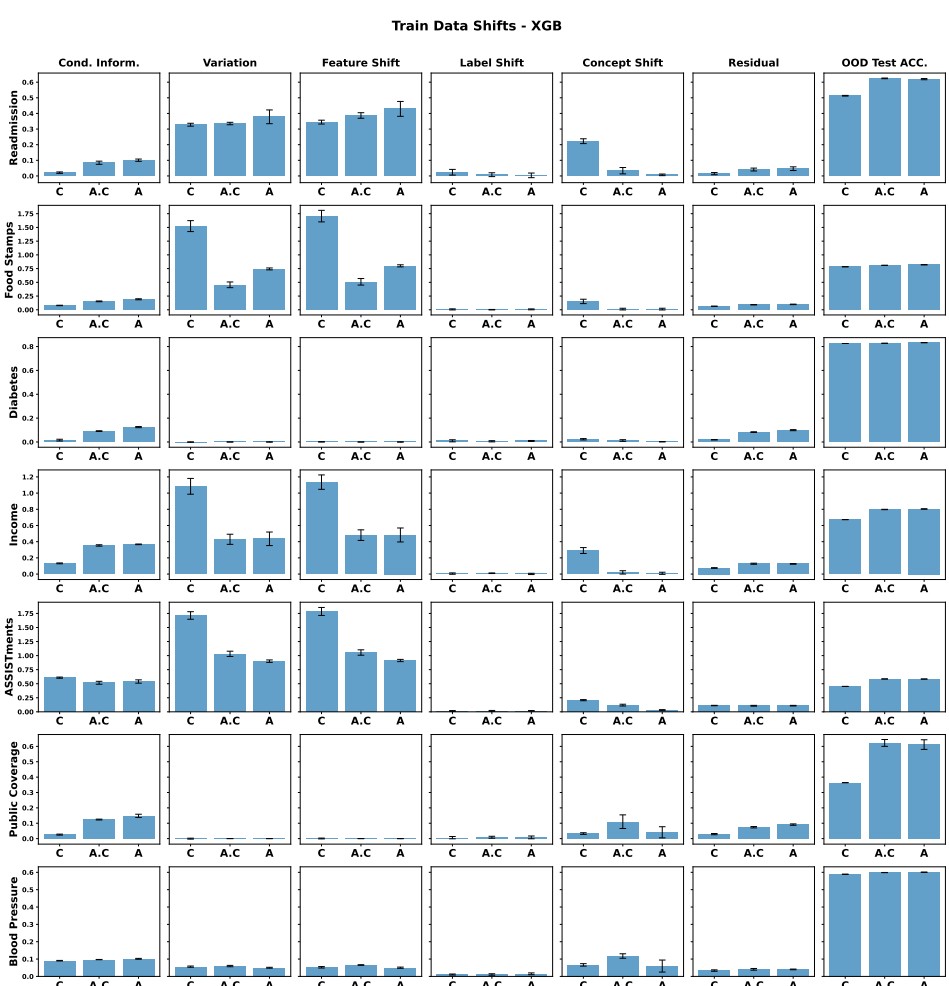

Figure C6: Decomposition of information metrics on train data for XGBoost and groups of features: (C) Causal, (A.C) Arguably causal, (A) All.

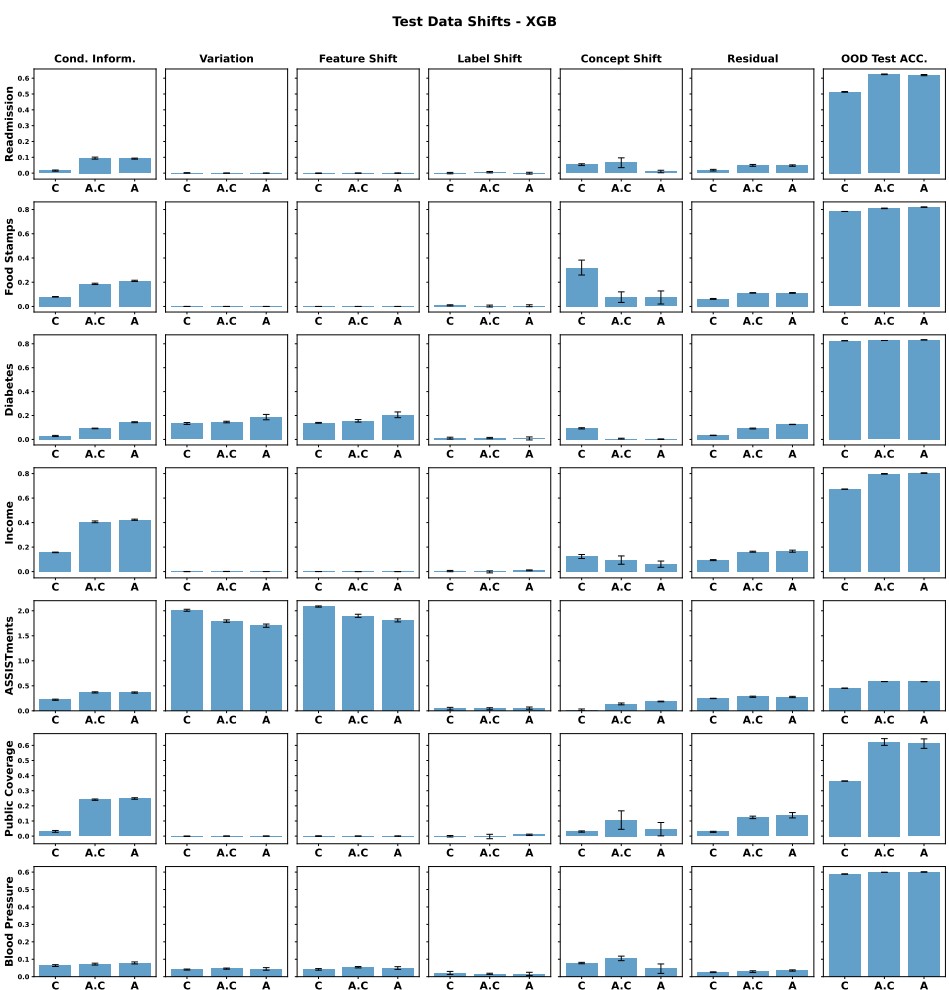

Figure C7: Decomposition of information metrics on test data for XGBoost and groups of features: (C) Causal, (A.C) Arguably causal, (A) All.

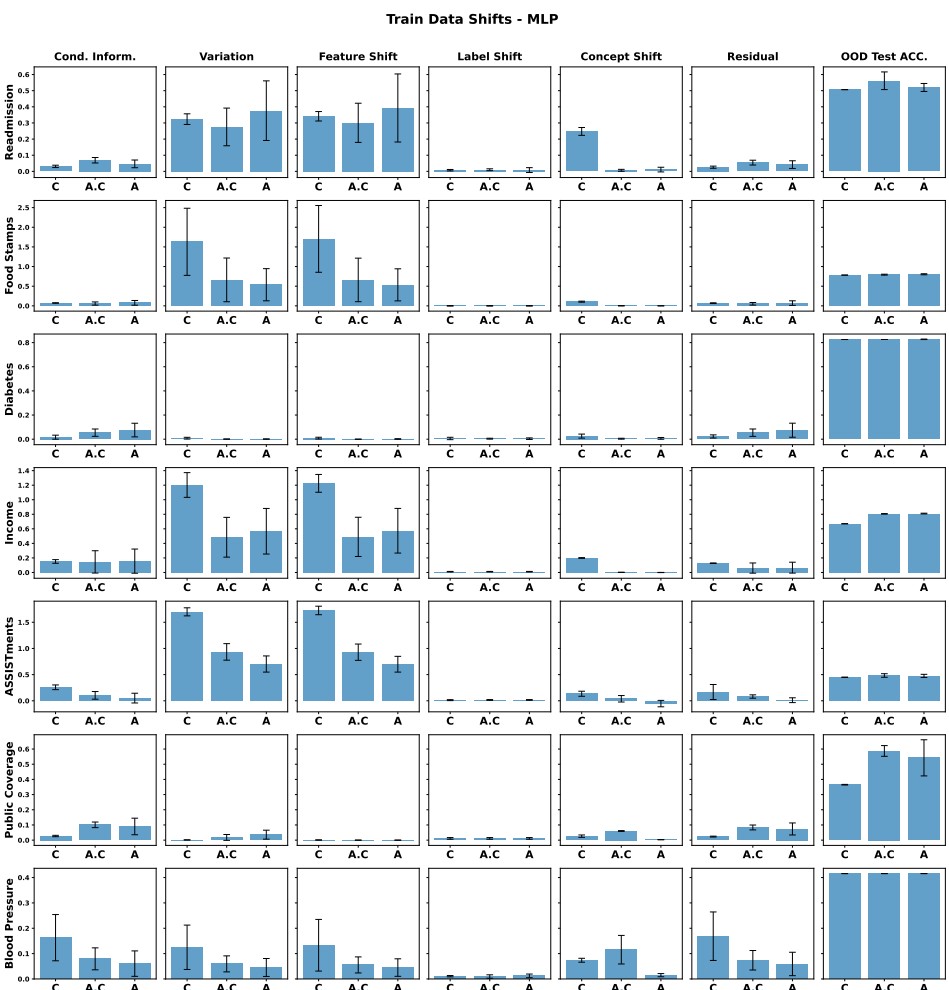

Figure C8: Decomposition of information metrics on train data for MLP and groups of features: (C) Causal, (A.C) Arguably causal, (A) All.

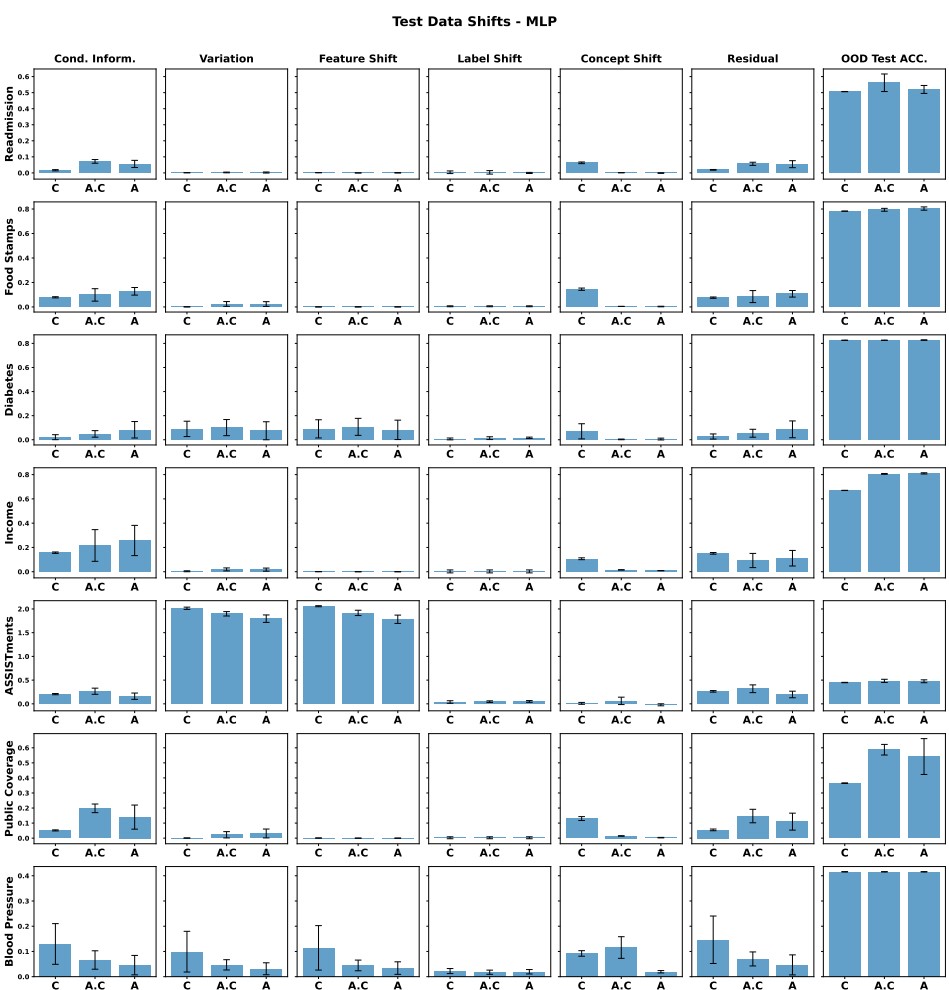

Figure C9: Decomposition of information metrics on test data for MLP and groups of features: (C) Causal, (A.C) Arguably causal, (A) All.

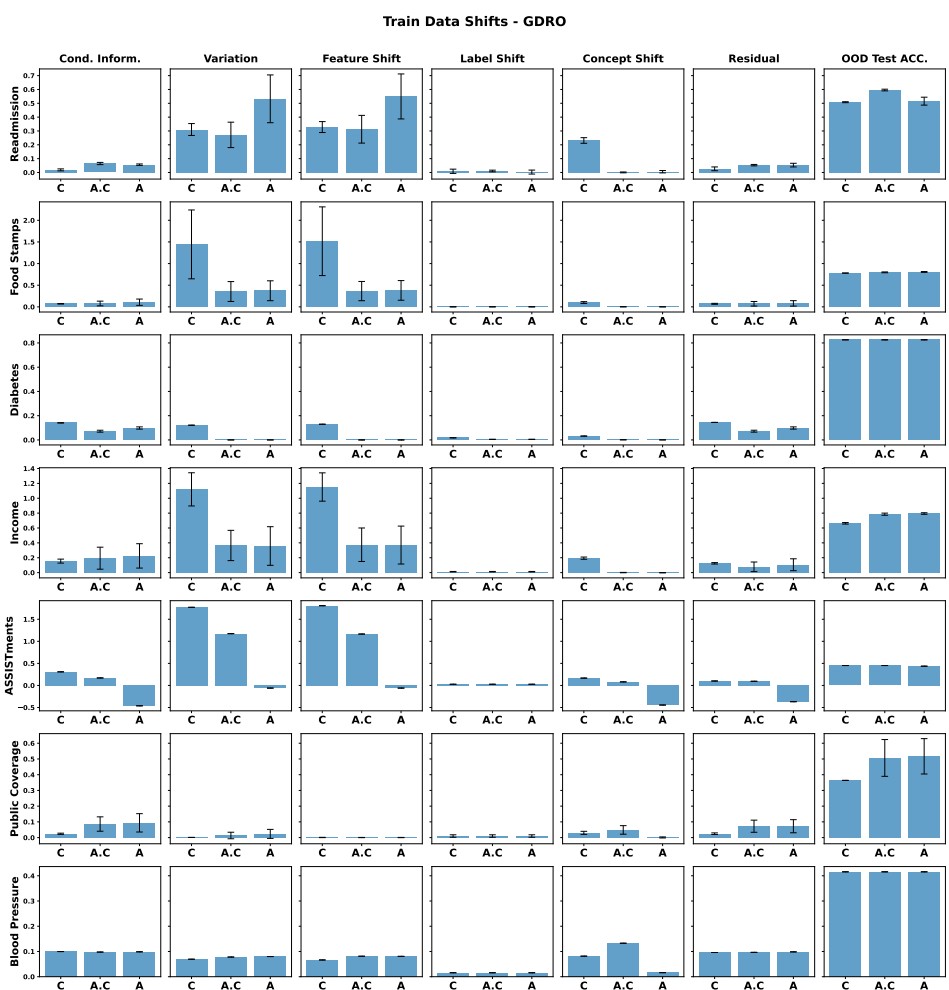

Figure C10: Decomposition of information metrics on train data for GDRO and groups of features: (C) Causal, (A.C) Arguably causal, (A) All.

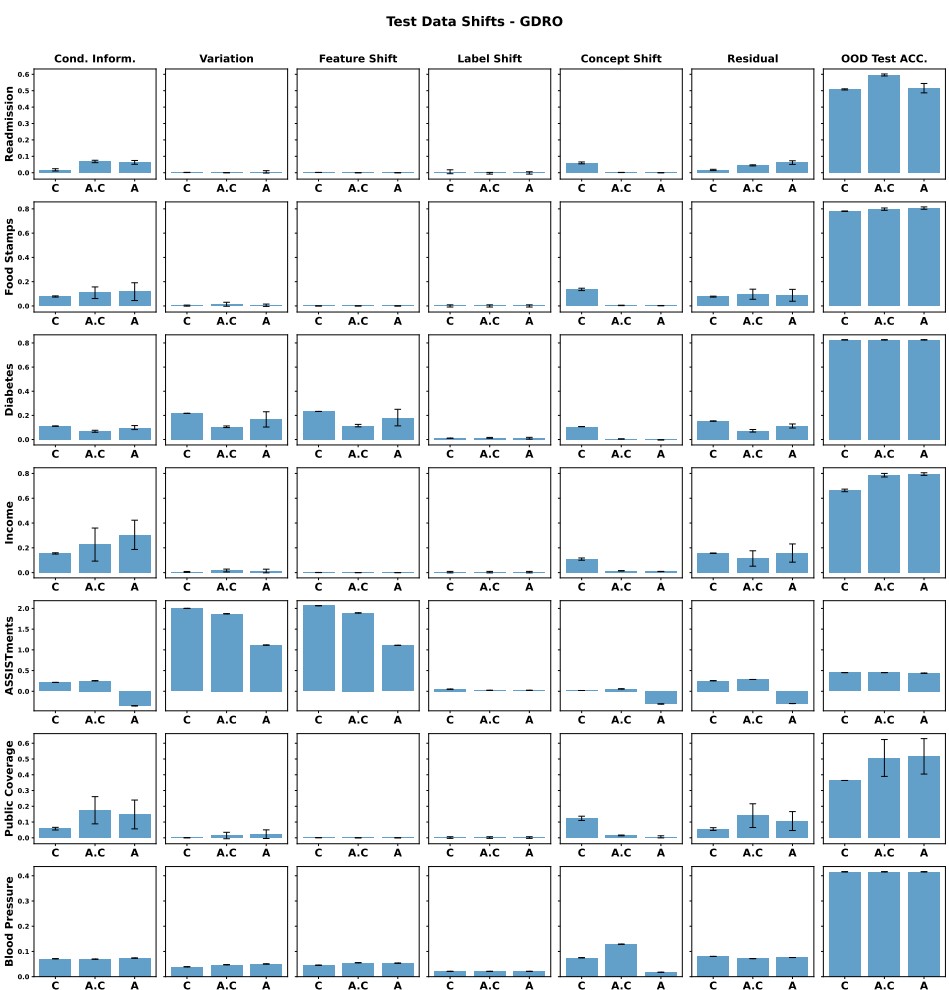

Figure C11: Decomposition of information metrics on test data for GDRO and groups of features: (C) Causal, (A.C) Arguably causal, (A) All.

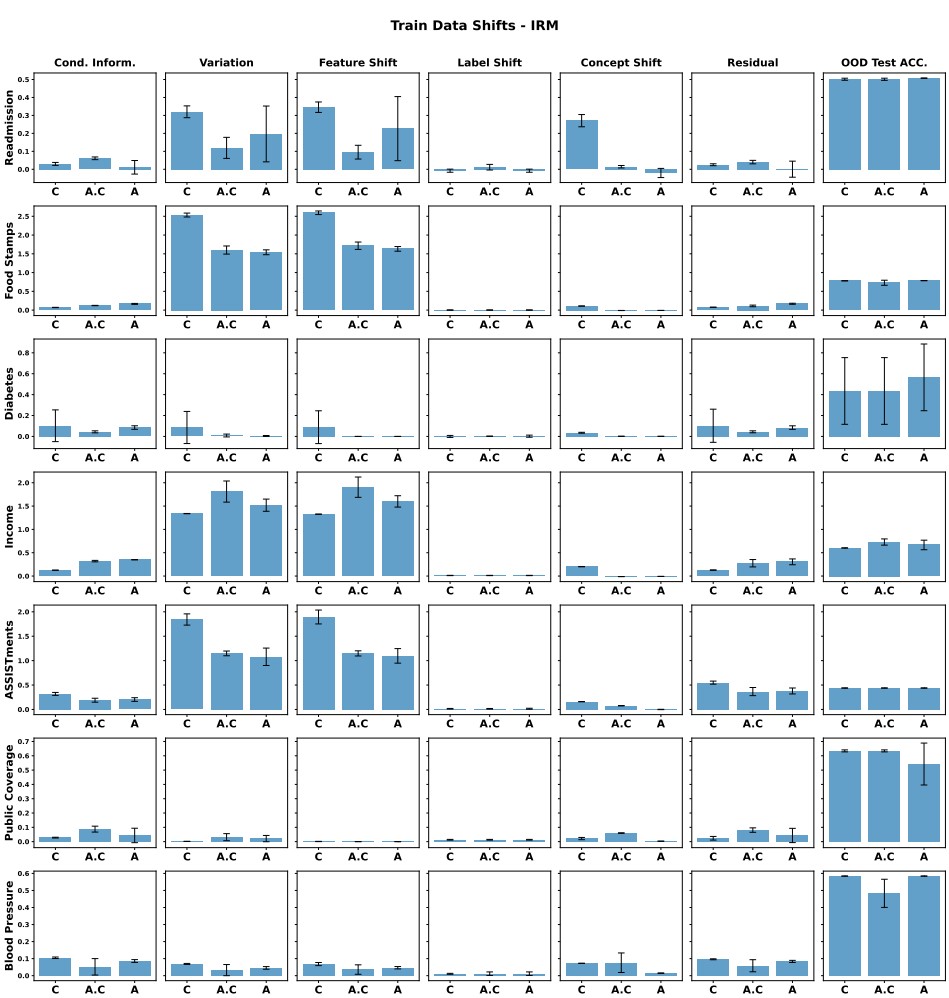

Figure C12: Decomposition of information metrics on train data for IRM and groups of features: (C) Causal, (A.C) Arguably causal, (A) All.

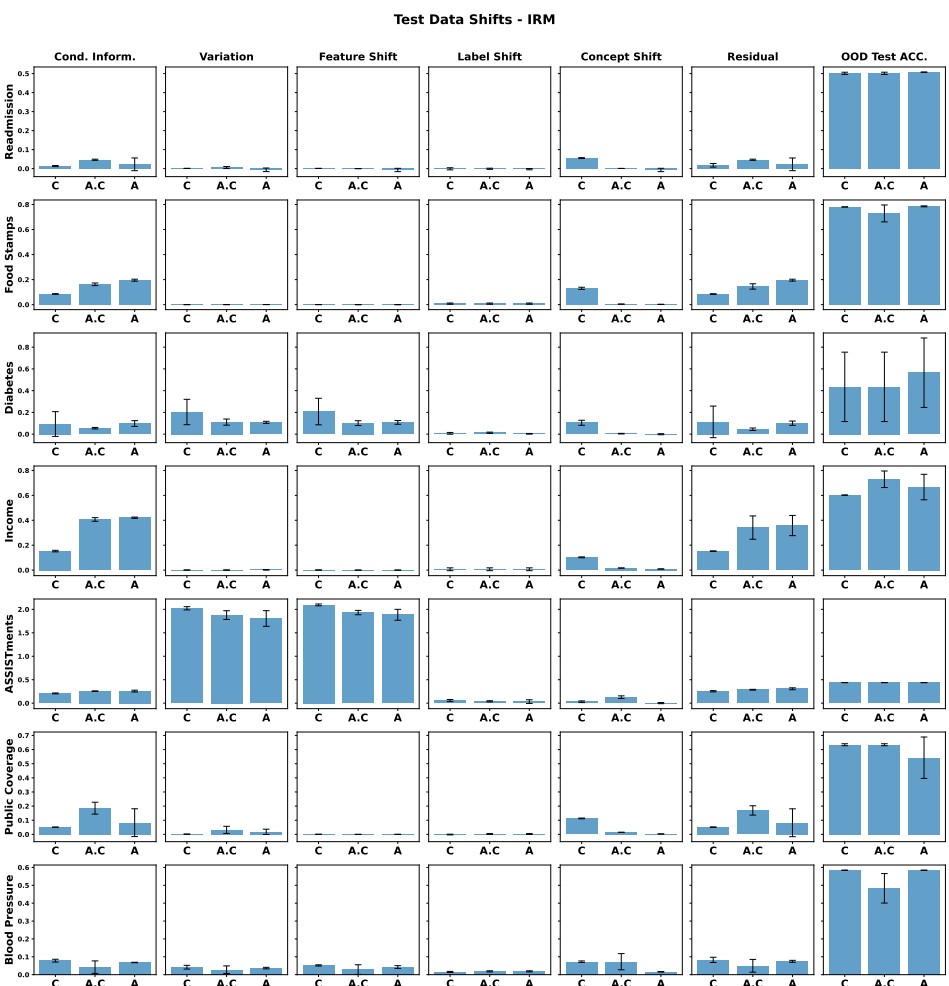

Figure C13: Decomposition of information metrics on test data for IRM and groups of features: (C) Causal, (A.C) Arguably causal, (A) All.

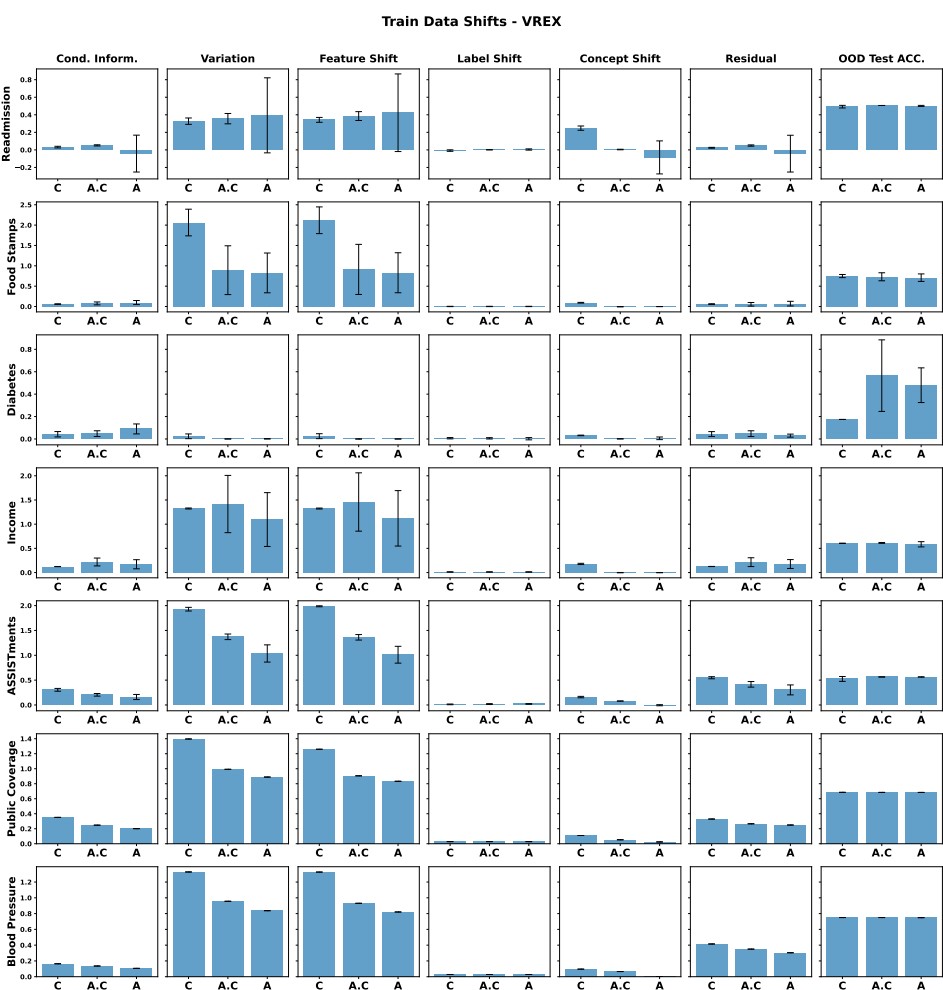

Figure C14: Decomposition of information metrics on train data for VREX and groups of features: (C) Causal, (A.C) Arguably causal, (A) All.

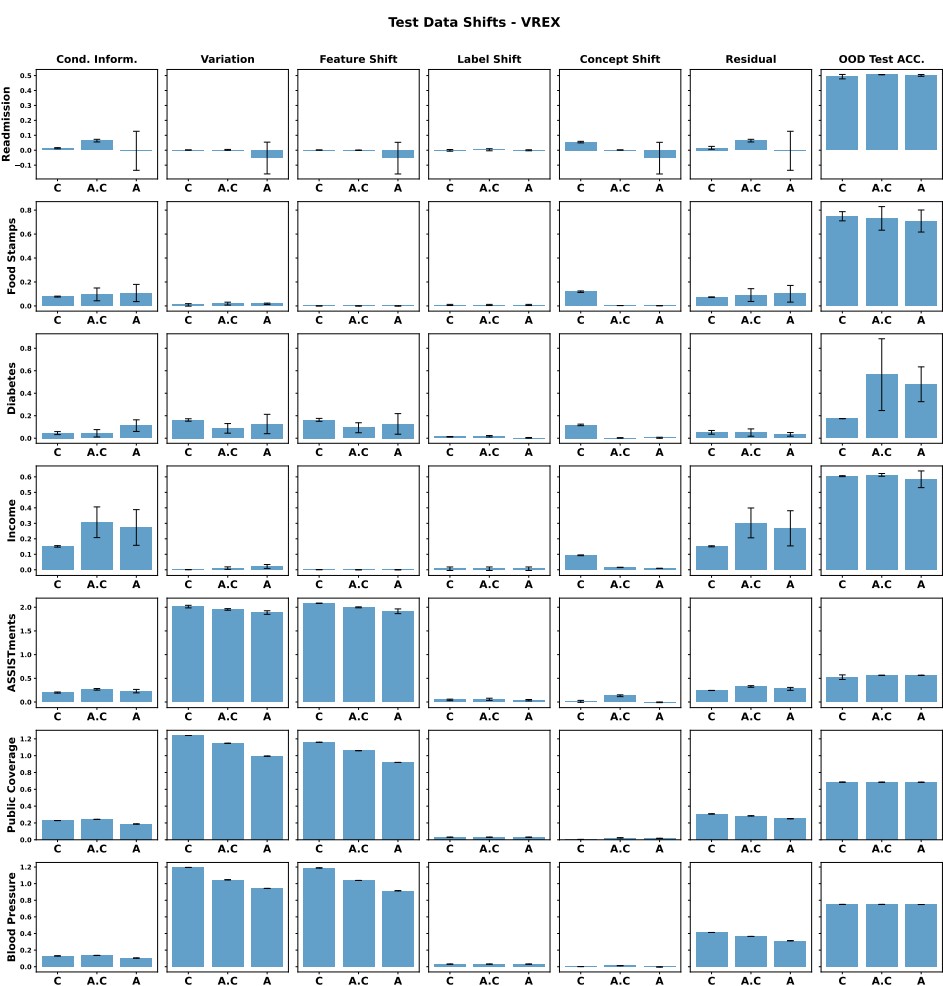

Figure C15: Decomposition of information metrics on test data for VREX and groups of features: (C) Causal, (A.C) Arguably causal, (A) All.

# D    RESULTS ON SYNTHETIC DATASETS

We conduct several experiments on synthetic datasets to investigate: (i) the impact of different informativeness criteria, (ii) how the level of confounder overlap across environments affects model performance, and (iii) the influence of the number of causal versus anti-causal variables on the performance. The data-generating process follows the structural equations in 12, with the causal structure $U \rightarrow X$, $U \rightarrow Y$, $U \rightarrow \mathbf{X}_I$, and $X \rightarrow Y$, where $U$ is an unobserved confounder, $X$ is an observed covariate, $Y$ is the target variable, and $\mathbf{X}_I$ represents additional informative covariates.

$$U \sim \mathcal{N}(\mu_u^e, \sigma_u) \qquad X \leftarrow f_X(U) + \mathcal{N}(\mu_x, \sigma_x)$$
$$X_i \leftarrow f_i(U) + \mathcal{N}(\mu_i, \sigma_i);\ X_i \in \mathbf{X}_I \qquad Y \leftarrow f_Y(X, U) + \mathcal{N}(\mu_y, \sigma_y)$$
$$(12)$$

We explore different functional forms for $f_X$, $f_i$, and $f_Y$, with the corresponding results presented in our analysis. Domain shifts are induced by systematically varying the environment-specific mean parameter $\mu_u^e$ of the confounder distribution. This allows us to examine how different degrees of distributional shift affect model performance across environments.

**Informativeness vs. accuracy:** In this set of experiments, we begin with linear structural equations: $f_X = 0.3U$, $f_i = 0.1U$, and $f_Y = X - 2U$. For these experiments $\mu_u^e \in \{-2, 2\}$ for training environments and $\mu_u^e \in \{0, 4\}$ for test environments. For the results presented in Figure 4 of the main paper, we consider one hidden confounding variable with $|\mathbf{X}_I| = 20$ informative covariates. As we increase the number of informative covariates from 0 to 20, we observe: (i) a reduction in mean squared error, (ii) improved conditional informativeness, (iii) enhanced feature shift, while (iv) decreased concept shift. We extend these findings by examining two additional informativeness criteria. First, we analyze the case with one hidden confounder and a single informative covariate ($|\mathbf{X}_I| = 1$) across varying noise levels $\sigma_i \in \{0, 0.1, \dots, 2.0\}$. Figure D16 demonstrates that as noise decreases from 2.0 to 0.0, we observe: (i) reduced mean squared error, (ii) improved conditional informativeness, (iii) enhanced feature shift, alongside (iv) decreased concept shift. Consistent with other synthetic experiments, the variation term remains zero throughout. Finally, we investigate a setting with 20 hidden confounders and corresponding 20 informative covariates, using modified structural equations $f_X = 0.3U$, $f_i = 0.2U$, and $f_Y = X - 2U$. Figure D17 shows that as we introduce informative covariates corresponding to hidden confounders, we again observe: (i) reduced mean squared error, (ii) improved conditional informativeness, (iii) enhanced feature shift, while (iv) decreased concept shift. These experiments demonstrate how different informativeness criteria help in achieving OOD generalization.

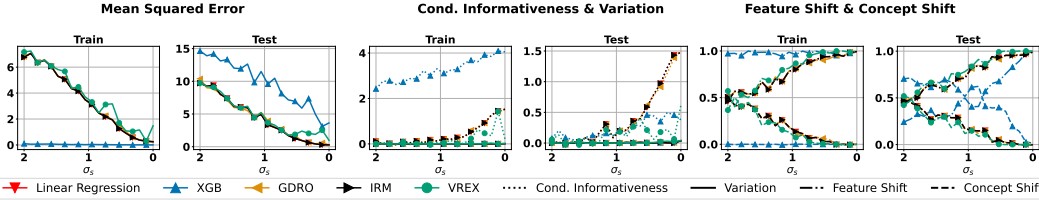

Figure D16: Adding proxy variable with low noise helps in reducing MSE, increasing conditional informativeness and feature shift while reducing concept shift.

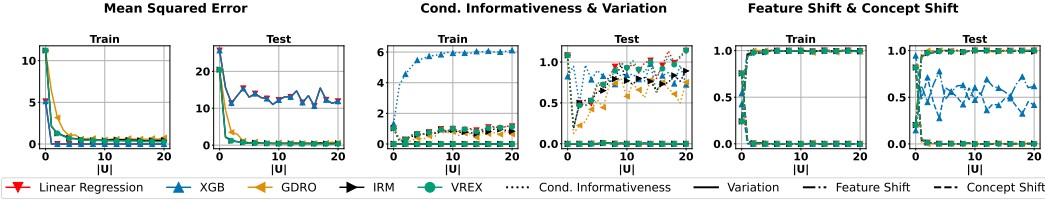

Figure D17: Adding more proxy variables $\mathbf{X}_I$ of a set of hidden confounding variables $\mathbf{U}$ that are informative to $Y$ helps in reducing MSE, increasing conditional informativeness and feature shift while reducing concept shift.

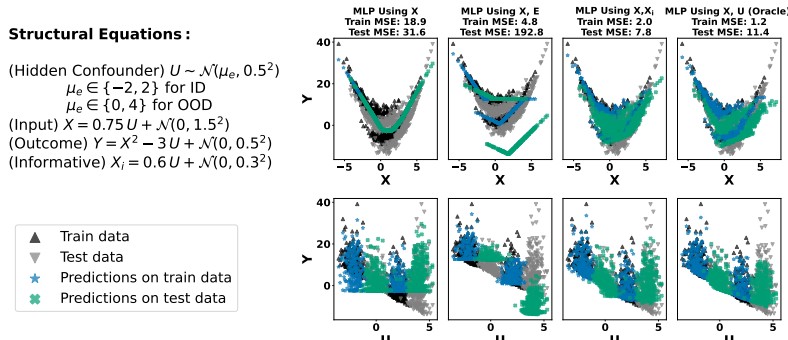

Figure D18: Performance of MLP on the low overlap non-linear setting, for sets of features: (i) $X$ (ii) $X$ and $E$ (environment statistics) (iii) $X$ and $X_i$ (iv) $X$ and $U$ (Oracle).

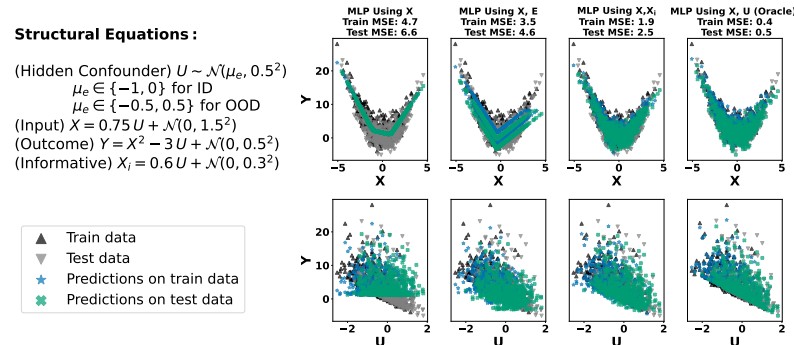

Figure D19: Performance of MLP on the high overlap non-linear setting, for sets of features: (i) $X$ (ii) $X$ and $E$ (environment statistics) (iii) $X$ and $X_i$ (iv) $X$ and $U$ (Oracle).

**Overlapping confounder support vs. accuracy:** As established in the main paper, recent work by Prashant et al. (2025) proposed an OOD generalization method for hidden confounding shift that assumes test confounder support remains within training support. While our results demonstrated successful learning of correct relationships in synthetic linear data when sufficient information about hidden confounders exists in observed covariates, we now extend this analysis to nonlinear data using MLP and XGB models under both low and high confounding support conditions (Figures D18–D21). In low confounding overlap settings (characterized by distant $\mu_e$ between ID and OOD data), models relying solely on $X$ learn an inadequate global function that fails to distinguish environments. Performance improves when incorporating environment-specific summary statistics of observed covariates, which helps capture environment-specific relationships, but the most significant gains occur only when leveraging additional informative covariates, underscoring their importance. For completeness, we include both oracle model results and $U$-$Y$ relationship scatter plots to elucidate the learned input-output mappings. These experiments confirm better performance under high confounding overlap as expected, but more importantly, reveal promising results in the low-overlap regime - a previously unstudied scenario that challenges the common support assumption in the literature.

**Number of causal vs anti-causal variables:** Recent benchmarks (Gardner et al., 2023) show that baseline methods (e.g., XGB, MLP) consistently match or outperform OOD generalization techniques (e.g., GDRO, IRM, VREX) in real-world datasets, coinciding with the prevalent causal structure $\mathbf{X} \rightarrow Y$ in tabular data. To test if baseline superiority stems from the causal structure: $\mathbf{X} \rightarrow Y$, we vary the causal-to-anti-causal ratio $\rho = |\mathbf{X}_C|/|\mathbf{X}_A| \in \{0.0, \ldots, 1.0\}$ with $|\mathbf{X}| = 50$ and $\mathbf{X} = \mathbf{X}_C \cup \mathbf{X}_A$ (e.g., $\rho = 0.5$ gives $|\mathbf{X}_C| = |\mathbf{X}_A| = 25$). Results (Figure D22) show baseline resilience even under the causal structures $\mathbf{X}_C \rightarrow Y \rightarrow \mathbf{X}_A$, though no method dominates universally (Figures D23, D24). Baselines achieve superior PMA-OOD scores (Gardner et al., 2023) (fraction of maximum OOD accuracy). We evaluate: (1) ERM-based methods (XGBoost (Chen & Guestrin, 2016), LightGBM (Ke et al., 2017), MLP, ResNet (Gorishniy et al., 2021), SAINT (Somepalli et al.,

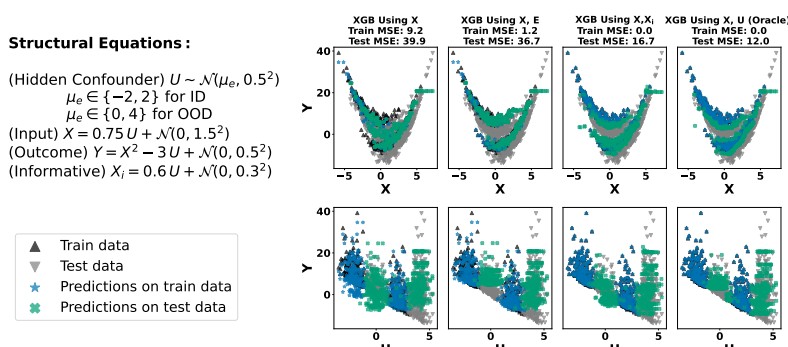

Figure D20: Performance of XGB on the low overlap non-linear setting, for sets of features: (i) $X$ (ii) $X$ and $E$ (environment statistics) (iii) $X$ and $X_i$ (iv) $X$ and $U$ (Oracle).

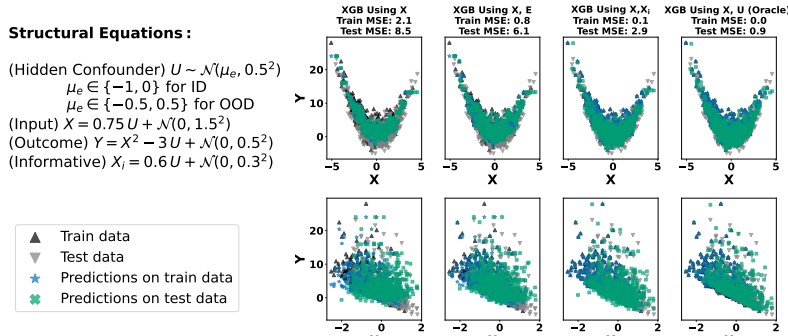

Figure D21: Performance of XGB on the low overlap non-linear setting, for sets of features: (i) $X$ (ii) $X$ and $E$ (environment statistics) (iii) $X$ and $X_i$ (iv) $X$ and $U$ (Oracle).

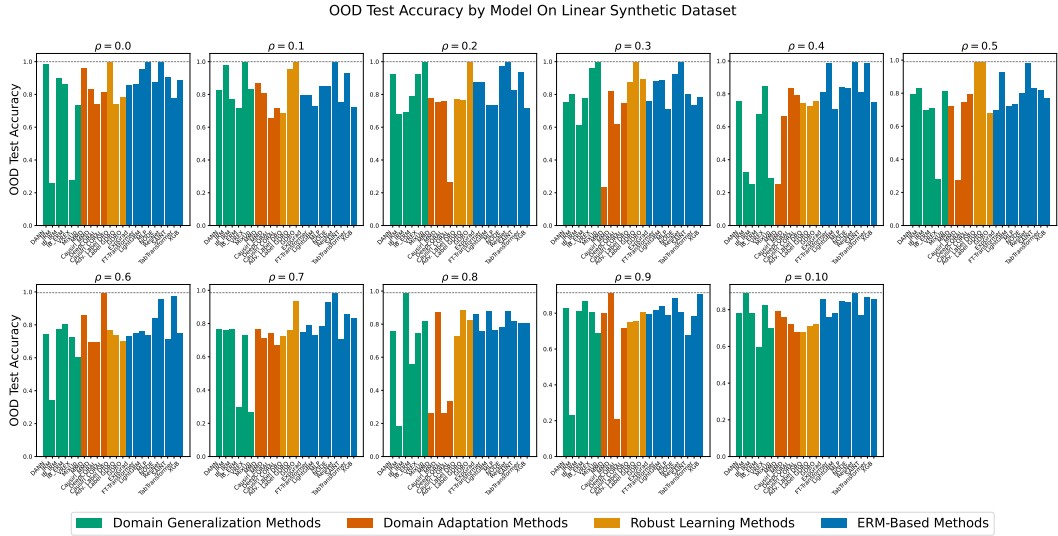

Figure D23: OOD test accuracy on the linear setting per model for different values of $\rho$.

2021), TabTransformer, NODE (Popov et al., 2019), FT-Transformer, ExpGrad (Agarwal et al., 2018)); (2) OOD generalization methods (IRM (Arjovsky et al., 2019), IB-IRM, IB-ERM (Gulrajani & Lopez-Paz, 2021), CausIRL (Chevalley et al., 2022), DANN (Ajakan et al., 2014), MMD (Li et al., 2018), DeepCORAL (Sun & Saenko, 2016a), V-REx (Krueger et al., 2021), Domain Mixup (Xu et al., 2020; Yan et al., 2020)); and (3) Robust optimization methods (DRO (Levy et al., 2020), GroupDRO (Sagawa et al., 2019), Label DRO, Adversarial Label DRO (Zhang et al., 2020)).

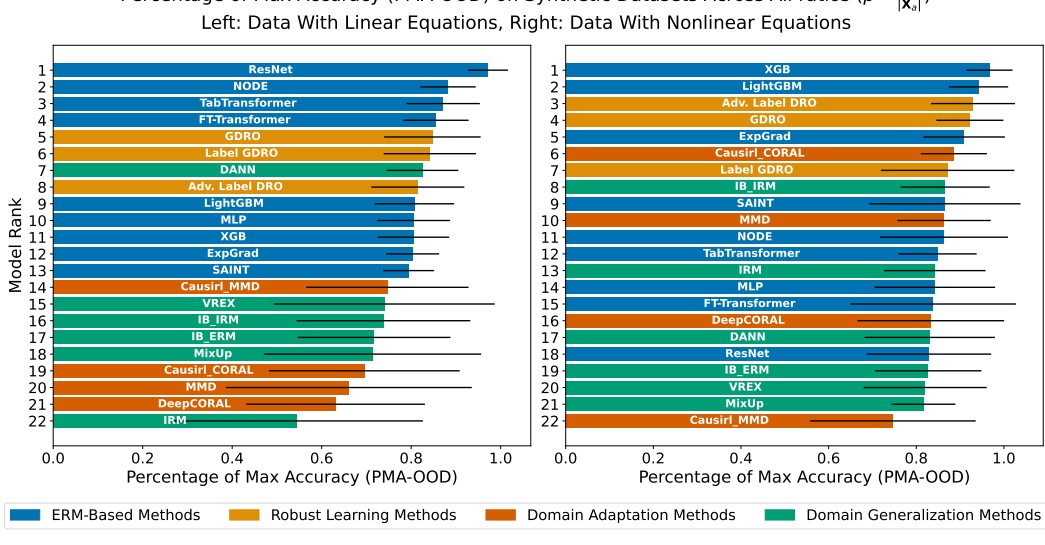

Figure D22: ERM-based methods achieve a better percentage of maximum OOD (PMA-OOD) test accuracy on the synthetic datasets, in line with the real-world results of (Gardner et al., 2023).

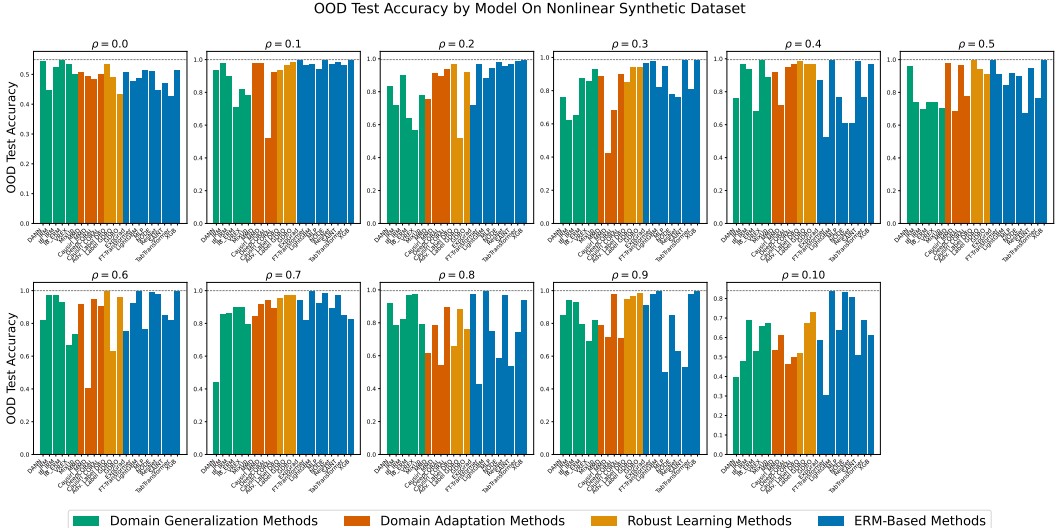

Figure D24: OOD test accuracy on the non-linear setting per model for different values of $\rho$

# E  QUALITATIVE ANALYSIS OF HIDDEN CONFOUNDING SHIFT IN REAL-WORLD DATASETS

In this section, we qualitatively study the presence of hidden confounding shifts in real-world datasets. Tables E3-E18 present potential hidden confounders influencing various observed covariates in several real-world datasets (Nastl & Hardt, 2024). For additional details regarding these datasets, we refer to (Gardner et al., 2023; Nastl & Hardt, 2024). We use GPT4o (Achiam et al., 2023) for this task, and we acknowledge that the list of unobserved confounding variables provided is not exhaustive. This study supports the arguments for the existence and impact of hidden confounding shifts across domains in real-world datasets. These GPT-4o responses are solely meant for semantic insight and are not implicitly or explicitly used in other experiments.

**Prompt:**  We use the prompt below to query GPT4o to get possible hidden confounders.

```
    For a target variable, you will be given lists of causal,
arguably causal, anti-causal, and spurious covariates.  Also,
you will be provided with environment variables such that
different datasets are collected in environments induced by these
environment variables.  Provide possible hidden confounders that
cause any of the target, causal, arguably causal, spurious, and
anti-causal covariates.  That is, the hidden confounders should
influence the distribution of their children in different domains.
Ensure hidden confounders are not a part of the given variables,
and your explanation should include how hidden confounders
influence their children in different environments.
```

Table E3: **Dataset:** Food stamps. **Target:** Food stamp recipiency in the past year for households with child. **Environments:** regions in the United States.

| Hidden confounder | Affected variables | Reason for confounding |
|---|---|---|
| Local economic conditions | Food stamp recipiency, Age, Sex, Race, Marital status | Poor economic conditions can increase food insecurity (raising food stamp usage), while also influencing population demographics due to migration, employment patterns, and household formation. |
| State or regional public assistance policies | Food stamp recipiency, Marital status, Number of children, Household income | Generous or restrictive assistance policies directly affect food stamp eligibility and indirectly influence household structure and income distribution. |
| Cultural attitudes towards welfare and public assistance | Food stamp recipiency, Marital status, Disability, Race, Ethnicity | Social stigma or support for welfare affects both the uptake of food stamps and social norms around marriage, disability reporting, and identity categories. |
| Regional health care conditions | Food stamp recipiency, Disability, Cognitive difficulty, Hearing difficulty, Vision difficulty | Regions with limited healthcare access may show higher disability prevalence and greater reliance on food stamps due to increased financial and care burdens. |

Table E4: **Dataset:** Income. **Target:** Total person's income $\geq 56K$ for employed adults. **Environments:** regions in the United States.

| Hidden confounder | Affected variables | Reason for confounding |
|---|---|---|
| Local economic conditions | Income, Occupation, Educational attainment, Marital status | Stronger local economies offer better-paying jobs and education access, increasing income; they also influence occupational choices and household formation patterns. |
| State or regional tax and labor policies | Income, Class of worker, Educational attainment, Marital status | Tax incentives and labor protections affect wages and employment types, while also shaping decisions about education and family due to financial security. |
| Cost of living and regional affordability | Income, Occupation, Marital status, Educational attainment | Higher living costs necessitate higher incomes and may drive occupational or educational shifts; they also influence marriage or cohabitation decisions. |
| Cultural norms and regional economic history | Income, Educational attainment, Occupation | Cultural values and historical industry presence shape education levels and job opportunities, which in turn affect income distributions. |

Table E5: **Dataset:** Public coverage. **Target:** Coverage of non-Medicare eligible low-income individuals. **Environments:** Disability statuses.

| Hidden confounder | Affected variables | Reason for confounding |
|---|---|---|
| State or regional healthcare policies | Public health coverage, Marital status, Employment status, Income | Differences in Medicaid expansion and public insurance eligibility influence health coverage; these policies also affect economic stability, employment, and family dynamics. |
| Economic conditions and poverty levels | Public health coverage, Employment status, Income, Marital status | Poorer regions tend to have lower employment rates and incomes, which increase reliance on public health coverage and influence marriage and household composition. |
| Cultural attitudes towards disability and healthcare | Public health coverage, Disability status, Cognitive difficulty, Hearing difficulty, Vision difficulty | Stigma or support for disability and public care varies culturally, affecting both the reporting of disabilities and the likelihood of seeking or receiving public coverage. |
| Urban vs. rural divide | Public health coverage, Educational attainment, Marital status, Occupation | Urban areas typically offer better healthcare access, education, and jobs, all of which affect coverage likelihood and socioeconomic indicators. |

Table E6: **Dataset:** Unemployment. **Target:** Classify whether a person is unemployed. **Environments:** Educational attainments.

| Hidden confounder | Affected variables | Reason for confounding |
|---|---|---|
| Local economic conditions | Employment status, Occupation, Marital status, Mobility status | Regional economic strength affects job availability and unemployment rates; it also shapes occupation types, migration decisions, and household stability. |
| State or regional labor market policies | Employment status, Occupation, Educational attainment, Marital status | Labor regulations and unemployment benefits vary by region, affecting hiring practices, education incentives, and family structures. |
| Cultural and social norms around employment | Employment status, Disability status, Marital status | Cultural attitudes toward work and dependency influence unemployment reporting and societal roles around disability and family responsibilities. |
| Health and disability status | Employment status, Disability status, Cognitive difficulty, Hearing difficulty, Vision difficulty | Poor health or disabilities reduce the ability to work, directly increasing unemployment and shaping associated health-related variables. |

Table E7: **Dataset:** Voting. **Target:** Classify whether a person voted in the U.S. presidential election. **Environments:** United States census regions.

| Hidden confounder | Affected variables | Reason for confounding |
|---|---|---|
| Political, family, and peer influences | Voted in national election, Party identification, Political participation | Social networks shape political ideology and engagement, influencing both voting likelihood and party alignment. |
| Media consumption habits | Voted in national election, Party identification, Political knowledge, Voting behavior | Media exposure affects awareness of political issues and biases, influencing party affiliation, political knowledge, and voting participation. |
| Social capital | Voted in national election, Political participation, Interest in elections | Strong community ties and civic networks increase political interest and participation, leading to higher voter turnout. |
| Civic education and political engagement programs | Voted in national election, Interest in elections, Political knowledge | Educational programs raise political awareness and civic responsibility, influencing both knowledge levels and voting decisions. |
| Historical and cultural context | Voted in national election, Party identification, Interest in elections | Historical events and regional political culture affect interest in elections and party alignment, which in turn influence voting behavior. |

Table E8: **Dataset:** Hypertension. **Target:** Whether a person has hypertension. **Environments:** Body Mass Index (BMI) values.

| Hidden confounder | Affected variables | Reason for confounding |
|---|---|---|
| Socioeconomic status | Income, Employment status, Smoking habits, Alcohol consumption, Physical activity, Healthcare access, Medical cost, High blood pressure diagnosis | Lower socioeconomic status reduces access to healthcare and healthy lifestyle options, leading to poor diet, limited activity, and delayed diagnosis, which jointly influence both BMI and hypertension risk. |
| Access to healthcare services | Healthcare access, Medical costs, Smoking habits, Alcohol consumption, Physical activity, High blood pressure diagnosis | Limited healthcare access results in under-diagnosis and unmanaged hypertension, while also affecting lifestyle choices that vary with BMI, confounding the relationship between BMI and hypertension. |
| Genetic predisposition | Age group, Race, Sex, Smoking habits, Diabetes, High blood pressure diagnosis | Genetic risk factors for hypertension may co-vary with demographic attributes and influence both hypertension prevalence and BMI distribution across subpopulations. |
| Psychosocial stress | Smoking habits, Alcohol consumption, Physical activity, High blood pressure diagnosis, Diabetes, Age group | Chronic stress alters behavior (e.g., smoking, inactivity) and physiological responses, contributing to both increased BMI and elevated blood pressure, thus confounding the BMI–hypertension link. |
| Environmental factors | Physical activity, Diet, Smoking habits, High blood pressure diagnosis, Diabetes, BMI category | Living environments affect access to recreational spaces, food quality, and pollution exposure, influencing both BMI and hypertension risks. These vary across BMI categories, creating confounding. |
| Dietary habits beyond fruits and vegetables | Alcohol consumption, Smoking habits, Physical activity, High blood pressure diagnosis, BMI category | High-sodium or processed-food diets raise both BMI and hypertension risk. Variation in such unmeasured dietary habits across BMI categories creates spurious associations with hypertension. |

Table E9: **Dataset:** College Scorecard. **Target:** Predict completion rate for first-time, full-time students at four-year institutions. **Environments:** Based on Carnegie Classifications.

| Hidden confounder | Affected variables | Reason for confounding |
|---|---|---|
| Institutional funding and resources | Accreditor, Control of institution, Highest degree awarded, In-state tuition, Out-of-state tuition, Cost of attendance, SAT scores | Wealthier institutions can offer better academic support, facilities, and programs, leading to higher completion rates and more selective admission profiles. This varies across Carnegie classifications. |
| Regional socio-economic factors | Region, Poverty rate, Unemployment rate, SAT scores | Economic conditions across regions affect affordability, student preparedness, and institutional support levels, all influencing both enrollment outcomes and graduation likelihood. |
| Demographic factors | HBCU flag, Federal loan recipient rate, Pell grant recipient rate, ACT scores, Undergraduate enrollment | Student demographics shape financial aid needs, academic preparation, and graduation rates. Their effect differs by institution type and selectivity under Carnegie categories. |
| Community support and engagement | Distance-education flag, Federal loan recipient rate, Pell grant recipient rate, SAT scores, Undergraduate enrollment | Supportive institutional communities improve retention and completion. Variation in engagement across institution types and student aid profiles induces confounding. |
| Admission selectivity | Admission rate, SAT (reading/math) midpoints, ACT midpoint, Undergraduate enrollment | Selective admissions correlate with better-prepared students and higher completion rates, and vary with institutional prestige and classification. |
| State and local policies | Region, Poverty rate, Unemployment rate, Cost of attendance | Differences in education funding and public policy affect cost structures and completion outcomes, interacting with institutional classification and regional demographics. |

Table E10: **Dataset:** ASSISTments. **Target:** Predict whether a student solves a problem correctly on the first attempt in an online learning tool. **Environments:** Different schools.

| Hidden confounder | Affected variables | Reason for confounding |
|---|---|---|
| Institutional teaching quality | Hint count, Attempt count, Skill ID, Problem type, Tutor mode, Position, Type, First action, Milliseconds to first response, Overlap time, Average confidence | Variation in instructional quality and pedagogy across schools affects how effectively students engage with content, leading to differences in problem-solving strategies, response behavior, and emotional states. |
| Student motivation | Hint count, Attempt count, Skill ID, Problem type, First action, Milliseconds to first response, Overlap time, Average confidence | Differences in intrinsic motivation across schools influence students' willingness to persevere, seek help, or give up quickly, affecting interaction and performance. |
| Classroom environment | Hint count, Attempt count, Tutor mode, Position, Type, First action, Average confidence | Peer dynamics, classroom culture, and noise levels vary by school and affect how confidently and independently students solve problems. |
| School technology infrastructure | Hint count, Attempt count, Tutor mode, Position, Type, First action, Milliseconds to first response, Overlap time, Average confidence | Access to reliable devices and fast internet differs by school, influencing response time, tool usage, and student experience. |
| Teacher-student interaction | Hint count, Attempt count, Tutor mode, Position, Type, First action, Milliseconds to first response, Overlap time, Average confidence | The level of teacher guidance and feedback shapes how much support students require during problem-solving, affecting engagement and confidence. |
| Previous academic performance | Hint count, Attempt count, Skill ID, Problem type, First action, Average confidence | Students' prior achievement affects how easily they solve problems, their need for assistance, and their confidence, all of which vary across schools. |

Table E11: **Dataset:** ICU. **Target:** Predict whether the patient will stay in the ICU for longer than 3 days. **Environments:** Insurance types.

| Hidden confounder | Affected variables | Reason for confounding |
|---|---|---|
| Socioeconomic status (SES) | Age, Gender, Ethnicity, Height , Weight , Bicarbonate, $CO_2$, $pCO_2$, $pO_2$, Lactate, Sodium, Hemoglobin, Oxygen saturation, Respiratory rate, etc. | SES influences access to healthcare, preventive services, and overall health status. Differences in SES across insurance types lead to variability in pre-ICU health, physiological indicators, and ICU stay duration. |
| Hospital resources and care quality | Age, Gender, Ethnicity, Height , Weight , Bicarbonate, $CO_2$, $pCO_2$, $pO_2$, Lactate, Sodium, Hemoglobin, Oxygen saturation, Respiratory rate, Heart rate, etc. | Hospital infrastructure and care standards affect monitoring, intervention speed, and clinical decisions. These factors vary by insurance coverage and influence ICU outcomes and vitals. |
| Comorbidities | Age, Gender, Ethnicity, Height , Weight , Bicarbonate, $CO_2$, $pCO_2$, $pO_2$, Lactate, Sodium, Hemoglobin, Oxygen saturation, Respiratory rate, Heart rate, etc. | Presence of chronic conditions (e.g., diabetes, cardiovascular disease) affects both the need for prolonged ICU care and physiological measurements. The distribution of comorbidities differs across insurance types. |
| Insurance-related treatment variability | Age, Gender, Ethnicity, Height , Weight , Bicarbonate, $CO_2$, $pCO_2$, $pO_2$, Lactate, Sodium, Hemoglobin, Oxygen saturation, Respiratory rate, etc. | Differences in treatment timing, intensity, and access to specialists based on insurance policies affect ICU stay duration and clinical metrics. |
| Genetic factors | Age, Gender, Ethnicity, Height , Weight , Bicarbonate, $CO_2$, $pCO_2$, $pO_2$, Lactate, Sodium, Hemoglobin, etc. | Inherited traits influence predisposition to organ failure, metabolic responses, and recovery trajectories. These effects are partially mediated by ethnicity and age distributions, which vary across insurance groups. |

Table E12: **Dataset:** Hospital mortality. **Target:** Classify whether an ICU patient expires in the hospital during their current visit. **Environments:** Insurance types.

| Hidden confounder | Affected variables | Reason for confounding |
|---|---|---|
| Socioeconomic status (SES) | Age, Gender, Ethnicity, Height, Weight, Bicarbonate, Lactate, Sodium, Hemoglobin, Oxygen saturation, Respiratory rate, Systolic blood pressure, White blood cell count, etc. | SES shapes access to timely and high-quality care, preventive services, and general health status. Patients with higher SES often have better insurance and outcomes, leading to confounding with mortality risk. |
| Comorbidities | Age, Gender, Ethnicity, Height, Weight, Bicarbonate, $CO_2$, $pCO_2$, $pO_2$, Lactate, Sodium, Hemoglobin, Oxygen saturation, Respiratory rate, Heart rate, Systolic blood pressure, etc. | Pre-existing conditions such as diabetes or heart disease increase mortality risk and influence physiological features. Their prevalence differs by insurance type, creating confounding. |
| Hospital resources and care quality | Age, Gender, Ethnicity, Height, Weight, Bicarbonate, $CO_2$, $pCO_2$, $pO_2$, Lactate, Sodium, Hemoglobin, Oxygen saturation, Respiratory rate, Heart rate, Systolic blood pressure, White blood cell count, etc. | Access to advanced treatments, trained staff, and timely interventions influences survival rates. These factors correlate with insurance coverage, confounding mortality outcomes. |
| Genetic factors | Age, Gender, Ethnicity, Height, Weight, Bicarbonate, $CO_2$, $pCO_2$, $pO_2$, Lactate, Sodium, Hemoglobin, Oxygen saturation, Respiratory rate, Heart rate, etc. | Genetic predispositions affect disease susceptibility and treatment responses. Variations in genetic risk factors may correlate with demographic traits across insurance types. |
| Lifestyle and behavioral factors | Age, Gender, Ethnicity, Height, Weight, Bicarbonate, $CO_2$, $pCO_2$, $pO_2$, Lactate, Sodium, Hemoglobin, Oxygen saturation, Respiratory rate, Heart rate, Temperature, Systolic blood pressure, White blood cell count, etc. | Behaviors such as smoking, diet, and physical activity affect long-term health and mortality risk. These behaviors vary systematically with SES and insurance coverage, influencing both target and physiological features. |

Table E13: **Dataset:** Childhood lead. **Target:** Predict blood lead levels above CDC blood level reference value. **Environments:** Poverty-income ratios.

| Hidden confounder | Affected variables | Reason for confounding |
|---|---|---|
| Environmental Exposure | Blood lead levels, Country of birth, Age, Race and Hispanic origin | Environmental exposure to lead influences blood lead levels, and this varies significantly across socio-economic groups. People in lower PIR groups are more likely to live in areas with higher lead contamination, which contributes to higher blood lead levels. Furthermore, environmental factors may affect the demographic distribution (e.g., country of birth, race). |
| Access to Healthcare | Blood lead levels, Age, Gender, Race and Hispanic origin, Marital status, Education | Limited access to healthcare, especially in lower PIR groups, means fewer opportunities for detection and treatment of lead poisoning. This results in higher blood lead levels, with disparities also influencing demographic variables like age, gender, and education. Additionally, healthcare access varies by insurance and socio-economic status, further confounding the relationships. |
| Diet and Nutrition | Blood lead levels, Age, Gender, Race and Hispanic origin | Dietary factors, such as poor nutrition in lower PIR groups, can exacerbate lead absorption. Malnutrition increases the body's susceptibility to lead poisoning, raising blood lead levels. In contrast, higher PIR groups may have better access to nutritious foods, lowering lead absorption, thus creating a confounding effect in how socio-economic status and race influence lead toxicity. |
| Housing Conditions | Blood lead levels, Country of birth, Race and Hispanic origin, Marital status | Older housing conditions, which are more prevalent in lower PIR groups, contribute significantly to elevated lead exposure (e.g., lead paint, poor plumbing). These living conditions directly influence blood lead levels and can also correlate with demographic factors like country of birth, race, and marital status. This introduces confounding, as socio-economic status impacts both exposure and the demographics of affected individuals. |
| Occupation | Blood lead levels, Age, Race and Hispanic origin, Marital status, Education | Certain occupations, which are more common among lower PIR groups, involve higher lead exposure (e.g., construction, manufacturing). Occupational lead exposure directly impacts blood lead levels and is often correlated with education, marital status, and socio-economic status. The varying prevalence of lead exposure by occupation introduces confounding, especially across different PIR groups. |

Table E14: **Dataset:** Diabetes. **Target:** Predict diabetes. **Environments:** Preferred race categories.

| Hidden confounder | Affected variables | Reason for confounding |
|---|---|---|
| Genetic predisposition | Diabetes, BMI, High blood pressure, High blood cholesterol | Genetic factors and family history contribute to both the onset of diabetes and comorbid conditions like obesity, hypertension, and high cholesterol. These genetic predispositions can make individuals more susceptible to diabetes, leading to confounding as they correlate with other health indicators. |
| Access to healthcare | Diabetes, Physical health, BMI, Healthcare coverage, Health checkups | Limited or unequal access to healthcare, especially in marginalized racial groups, leads to disparities in diabetes diagnosis, management, and comorbidity treatment. It also influences the frequency of health checkups and access to medications, which can confound the relationship between diabetes status and other health metrics. |
| Dietary habits and food availability | Diabetes, BMI, Physical health, Alcohol consumption, Fruit and vegetable intake | Dietary habits, often shaped by socioeconomic status and local food environments, influence weight, health behaviors (such as alcohol consumption), and diabetes risk. People in lower socioeconomic strata may have limited access to healthy food options, leading to higher BMI and increased diabetes risk, creating confounding effects on health outcomes. |
| Psychosocial stress and mental health factors | Diabetes, Mental health, Physical health, BMI, Physical activity, Doctor visits | Chronic stress and mental health issues, often higher in marginalized groups, contribute to diabetes development and complicate its management. These factors also affect physical health (e.g., weight gain due to stress) and health-seeking behaviors (e.g., fewer doctor visits), leading to confounding by influencing both diabetes risk and its associated variables. |
| Socioeconomic status beyond income | Diabetes, Income, Physical health, Healthcare coverage, Education level | Socioeconomic factors, such as occupation, education, and neighborhood wealth, influence access to healthcare, nutrition, and overall health behaviors. These factors can confound the relationships between diabetes and other socio-economic variables like income and education, as they shape opportunities for prevention and treatment. |

Table E15: **Dataset:** Sepsis. **Target:** Predict, from a set of fine-grained ICU data, whether a patient will experience sepsis onset within the next 6 hours. **Environments:** Lengths of ICU stay.

| Hidden confounder | Affected variables | Reason for confounding |
|---|---|---|
| Infection prevalence in ICU | SepsisLabel, Temperature (Temp), Leukocyte count (WBC), Heart rate (HR), Blood urea nitrogen (BUN) | Higher infection rates in certain ICU units can lead to a higher probability of sepsis onset (SepsisLabel). These infection rates influence biomarkers such as WBC, HR, and BUN, creating confounding because the unit's infection environment affects both the likelihood of sepsis and the observed clinical measures. |
| Quality of ICU Care | SepsisLabel, Fibrinogen concentration (Fibrinogen), Leukocyte count (WBC), Platelet count (Platelets) | Higher-quality care in certain ICUs may lead to earlier identification and treatment of sepsis, resulting in more accurate SepsisLabel predictions. Additionally, better care could affect biomarkers like fibrinogen, WBC, and platelets, which are critical in sepsis detection and progression, thereby confounding the relationships between these variables and the outcome. |
| Severity of underlying conditions | SepsisLabel, Blood urea nitrogen (BUN), Creatinine, Lactate, Calcium | Patients with severe chronic conditions (e.g., kidney disease, cardiovascular issues) are at a higher risk of sepsis and may show abnormal levels in biomarkers like BUN, creatinine, lactate, and calcium. These underlying conditions contribute to the SepsisLabel outcome and confound the relationship between the biomarkers and the likelihood of sepsis, varying across ICU units depending on patient population. |
| Patient's socio-economic status | Age (Age), Gender (Gender), Leukocyte count (WBC), Fibrinogen concentration (Fibrinogen), Platelet count (Platelets) | Socio-economic factors, such as access to healthcare, can influence both the likelihood of sepsis and the observed clinical biomarkers. For example, patients from lower socio-economic backgrounds may have delayed hospitalizations or inadequate care, which affects both SepsisLabel and the progression of sepsis as indicated by WBC, fibrinogen, and platelet levels. |
| Hospital-specific protocols and treatment guidelines | SepsisLabel, Lactate, Glucose, Creatinine | Differences in hospital protocols for sepsis treatment, such as timing of interventions and choice of sepsis bundles, can affect both the SepsisLabel and biomarkers like lactate, glucose, and creatinine. These protocols lead to variability in how sepsis is diagnosed and treated across different hospitals and ICU units, confounding the relationship between biomarkers and sepsis outcomes. |

Table E16: **Dataset:** Hospital readmission. **Target:** Predict whether a diabetic patient is readmitted to the hospital within 30 days of their initial release. **Environments:** Admission sources.

| Hidden confounder | Affected variables | Reason for confounding |
|---|---|---|
| Socio-economic status (SES) | Race, Gender, Age, Payer code, Medical specialty, Number of outpatient visits, Number of emergency visits, Number of inpatient visits, Diabetes medication prescribed | SES influences access to healthcare, patient demographics, and chronic disease rates. It can also determine the type of care received based on admission source (e.g., emergency department vs. outpatient settings). Differences in healthcare access, such as availability of medications, may impact readmission rates and associated variables like outpatient visits and prescribed medication. |
| Severity of illness | Primary diagnosis, Secondary diagnosis, Number of diagnoses, Discharge type, Medication changes (e.g., Insulin, Glipizide) | More severe illness increases the likelihood of readmission and influences the complexity of diagnoses and the treatments administered. The severity of illness may differ based on the admission source (e.g., emergency versus outpatient), impacting the number and type of diagnoses and treatments prescribed at discharge, affecting readmission likelihood. |
| Access to healthcare resources | Time in hospital, Discharge disposition, Number of procedures, Number of medications, Number of lab tests | Access to healthcare resources (e.g., time in hospital, availability of procedures and medications) influences treatment decisions and outcomes. Different admission sources may have varying levels of available resources, leading to different lengths of stay, the types of procedures performed, and overall treatment quality, which can affect the likelihood of readmission. |
| Patient's adherence to medication | Change in medications, Diabetes medication prescribed, Number of outpatient visits, Number of emergency visits | Adherence to prescribed medications is often influenced by socio-economic status, which can vary depending on admission source. Non-adherence may lead to medication changes, affecting diabetes control and subsequent readmission risk. The number of outpatient and emergency visits can also reflect how well a patient manages their diabetes and the likelihood of complications. |
| Hospital-specific protocols | Time in hospital, Number of diagnoses, Discharge disposition, Readmitted | Different hospitals and healthcare systems implement various protocols for discharge planning and readmission prevention, which can affect readmission rates. These protocols may vary by admission source, where patients admitted via the emergency department may receive different follow-up instructions and care than those admitted through other channels. |

Table E17: **Dataset:** MEPS. **Target:** Measure of health care utilization. **Environments:** Insurance types.

| Hidden confounder | Affected variables | Reason for confounding |
|---|---|---|
| Socioeconomic status (SES) | Years of education, Employment status, Hourly wage, Paid sick leave, Paid leave to visit doctor, Family size, Insurance coverage, Healthcare utilization | SES can influence access to healthcare services, insurance coverage, and the ability to use medical services. Insurance types often correlate with SES levels, where individuals with lower SES may be more likely to be insured by government programs (e.g., Medicaid), which in turn impacts healthcare utilization patterns across different SES groups. |
| Healthcare access | Paid sick leave, Insurance coverage, Employer offers health insurance, Healthcare utilization | Limited access to healthcare, such as lack of insurance or paid sick leave, directly impacts healthcare utilization. The type of insurance a person holds is often tied to access to various medical services. The level of coverage and accessibility differs across insurance types, influencing healthcare behaviors such as whether a patient can afford and utilize healthcare services. |
| Health behaviors | Perceived health status, Asthma medications, Limitations in physical functioning, Healthcare utilization | Lifestyle factors like smoking, alcohol consumption, and physical activity can directly affect health status and healthcare needs. Health behaviors differ across groups with different insurance types, and these behaviors contribute to healthcare utilization. Insurance coverage can also be influenced by perceived health status, which varies across insured groups, affecting utilization of medical services. |
| Chronic health conditions | Asthma medications, Perceived health status, Limitations in physical functioning, Healthcare utilization | Chronic conditions like asthma or diabetes increase the need for healthcare services, leading to higher healthcare utilization. People with chronic conditions are more likely to be covered by Medicare or Medicaid, which influences the type of insurance they hold and impacts their healthcare utilization and associated features (e.g., medications and physical limitations). |
| Regional healthcare infrastructure | Region, Family size, Healthcare utilization, Paid sick leave, Paid leave to visit doctor, Employment status | The quality and availability of healthcare infrastructure vary across regions, which can impact healthcare utilization. In underserved regions, people may be more reliant on public insurance options like Medicaid, affecting healthcare access and behaviors. The regional differences in healthcare systems can lead to disparities in access to medical services, insurance coverage, and utilization of healthcare. |

Table E18: **Dataset:** Poverty. **Target:** Predict household income-to-poverty ratio. **Environments:** Citizenship statuses.

| Hidden confounder | Affected variables | Reason for confounding |
|---|---|---|
| Social capital | Income, Unemployment compensation, Disability benefits, Family size, Housing conditions, Household income-to-poverty ratio | Stronger social support networks can improve access to income, benefits, and resources like unemployment or disability compensation. Social capital can vary based on citizenship status, which in turn influences income, housing conditions, and eligibility for benefits, confounding the relationship between household income and poverty ratios. |
| Workplace discrimination or bias | Household income, Worker's compensation, Unemployment compensation, Pension, Family income, Disability benefits | Discrimination in the workplace can limit opportunities for higher wages, pensions, or disability benefits, especially for marginalized groups (e.g., racial minorities, non-citizens). This bias varies by citizenship status and leads to biased associations between income, benefits, and poverty levels. |
| Healthcare access | Household income, Health insurance premiums, Medical expenses, Disability benefits, Family size, Health conditions, Medicaid/Medicare assistance | Limited access to healthcare, particularly for non-citizens, can lead to higher medical expenses and greater reliance on public health assistance. Citizenship status directly impacts eligibility for public healthcare programs, confounding relationships between medical expenses and income, and affecting poverty ratios. |
| Access to education and skills development | Educational attainment, Household income, Employment status, Unemployment compensation, Income from assistance, Savings | Disparities in educational opportunities, often linked to citizenship status, influence income potential and eligibility for government assistance. These disparities, in turn, affect the household income-to-poverty ratio and employment outcomes, creating confounding relationships between education and income. |
| Cultural factors | Family size, Household income, Savings, Disability benefits, Healthcare utilization, Income assistance, Living arrangements | Cultural factors influence financial management, family support, and the use of social programs. These factors can vary across citizenship statuses, leading to differences in how household income and assistance are distributed, and thus confounding the relationship between income, benefits, and financial behavior. |
| Housing market and rent conditions | Housing ownership, Housing costs, Family size, Household income, Rent payments, Social security benefits, Medical aid | Housing market conditions, especially rent disparities, can create financial strain and impact the household income-to-poverty ratio. Citizenship status often influences eligibility for housing assistance and rent subsidies, which confounds the relationship between housing costs and income, particularly in areas with significant immigrant populations. |

## LLM USAGE

In addition to querying LLMs about the semantic meaning of hidden confounding variables (§ E), LLMs are used to aid or polish writing.

