# OpenReview forum: "When Shift Happens - Confounding Is to Blame"
_ICLR.cc/2026/Conference — ICLR 2026 Poster_

### Official Review · Reviewer_ov4h · 2025-10-30

**Soundness:** 3
**Presentation:** 3
**Contribution:** 3
**Rating:** 6
**Confidence:** 3

**Summary:**

The paper investigates data generating processes in which confounding of $X$ and $Y$ takes place through an unobserved variable $U$ (‘hidden confounding’, Figure 3).

Theory:
Using mutual information (MI) of $Y$ and its prediction $\hat{Y}$ as objective (O) to be maximised, the authors provide results that show that in cases where either $X \to Y$ or $Y \to X$ (but not both), O is equal to the sum of the environment specific MI between $Y$ and learned representations $\phi(X)$, plus a residual term (will refer to this sum as S below), showing that in such scenarios learning environment specific information is expected to be superior to invariant approaches.

Experiments:
The empirical part of the paper focusses on the TableShift benchmark, which contains a range of tabular datasets with distribution shift. According to the authors, most of these datasets follow a data generating process in which either fully or predominantly $X \to Y$ (but not $Y \to X$)
Using measurements of MI, the authors find changes in conditional MI terms that would be expected under the data generating process the paper considers (Table 3). The authors further evaluate 5 different predictive methods (XGBoost, MLP, GDRO, IRM, VRex) on TableShift and find a positive spearman correlation between average test accuracy of a method and the value of S it achieves in both ID and OOD (Table 2).

**Strengths:**

- The fundamental problem of distribution shift is relevant and recent findings by Nasal and Hardt (2024) on the effectiveness of ERM make the paper timely
- The decomposition in Theorem 4.1 is innovative and a clever way to analyse different effects of hidden confounding shift.
- The derived result in Theorem 4.2 is insightful and relevant
- The experiments to corroborate the applicability of the assumptions in Propositions 4.1 and 4.2 are generally speaking well designed (but could be improved, see weaknesses)

**Weaknesses:**

- It would be good if the authors could show how their insights from Theorem 4.2 can be turned into practical algorithms. They mention ideas in the the paragraph “Remarks on conditional informativeness” on page 7, but do not pursue those to derive a method that maximises Eq (4) that is compared to existing methods in the experiment section. One idea could be for example to equip the standard MLP with environment variable(s) similar to Figure 2 (ii), e.g. the feature means or similar and see if this can improve results.
- Along similar lines, the empirical results in Table that are meant to corroborate practical relevance of Theorem 4.2, are all only observational and based on correlations of measured quantities. Doing some intervention on conditional informativeness - residual values of a method (e.g. actively minimising or maximising the term) and observing changes in accuracy would make a stronger point, or alternatively add some more methods (e.g. logistic regression ERM to keep it simple, or other invariant or domain generalisation approaches) or the same methods with different hyper parameters (e.g as found in Figure C4) to the analysis in Table 2 would make the correlation estimate more reliable.
- I think Eq (2) and (3) should be derived formally in a proposition, rather than justified in a few words of text as the derivation is not clear to me as such
- Estimating MI of continuous, high-dimensional variables is a non-trivial problem. It would be good to summarise  in the main text how this was achieved and reference where to look to find details on it in the Appendix


Minor:
- Presentation of results in Table 2: as is, it does not clearly show the correlation between conditional informativeness - residual and accuracy as the main focus is on the test accuracies and the spearman correlations are just somewhat floating around. Why not add the values of conditional informativeness - residual for both ID and OOD in the table to make the picture more complete? Or otherwise you might as well just give the Spearman correlation coefficients in line in the text, and don’t need to use all the space for the table.
- Proof of theorem 4.2 should be referred to in the main text.
- An example graph similar to Figure 3 (or an extension thereof) would be helpful to understand definition 4.3 / the concept of informative covariates.

**Questions:**

- Line 76: Why is learning an invariant representation sufficient? The paper seems to suggest the opposite in Eq 1?
- Line 209: Why is the sum over $\mathcal{R}^{\mathcal{D}^e}$ and not simply over $\mathcal{R}^{e}$? For the other methods in the same paragraph the objective is always stated using $\mathcal{R}^{e}$.
- Table 2: why are the authors excluding the results from Figure C4 in the analysis? Adding these would make the estimate of the correlation more robust (and also their value lower I think)

---

> ### Author Response · Authors · 2025-11-20
> **Response to Reviewer ov4h**
>
> We thank you for your insightful review. We will address your comments below.
>
> > Q1) It would be good if the authors could show how their insights from Theorem 4.2 can be turned into practical algorithms...
>
> We thank the reviewer for their insightful suggestions. Instead of directly maximizing the difference: conditional informativeness-residual, which is extremly time consuming to evaluate mutual information terms in each epoch, based on the insights from our paper, we suggest the following actionable algorithms, each of which could motivate a separate line of research that is beyond the scope of this paper.
>
> **Potential ideas for actionable algorithms:**
>
> * **Idea 1:** Based on the synthetic experiments in Figure 2, adding domain-specific information $\mathbf{X}_I$ that is informative about $Y$ can improve model performance. From the causal graph in Figure 3, such domain-specific signals are likely associated with the hidden confounder. Adding these variables increases conditional informativeness and reduces concept shift. This suggests that $\mathbb{P}(Y \mid \mathbf{X}, \mathbf{X}_I)$ is closer to invariant than $\mathbb{P}(Y \mid \mathbf{X})$. Designing a systematic informative-covariate augmentation pipeline, either by collecting extra features or by generating them, would provide immediate practical benefits.
>
>
> * **Idea 2:** From Theorem 4.2, minimizing the residual can also improve performance. Making the residual term $I(\phi(\mathbf{X}); Y \mid \hat{Y})$ zero is equivalent to enforcing the conditional independence $Y \perp \phi(\mathbf{X}) \mid \hat{Y}$. One approach to achieve this is imposing an adversarial conditional-independence penalty. Specifically, define a base model $\mathcal{B}$ that learns $\phi(\mathbf{X})$ and predicts $\hat{Y} = f(\phi(\mathbf{X}))$ by minimizing the usual cross-entropy loss $L_{\text{base}}$. Introduce an adversarial model $\mathcal{D}$ that predicts $Y$ from $(\phi(\mathbf{X}), \hat{Y})$ using its own cross-entropy loss $L_{\text{adv}}$. Train the models adversarially: $\mathcal{D}$ minimizes $L_{\text{adv}}$, while $\mathcal{B}$ minimizes $L_{\text{base}} - \lambda L_{\text{adv}}$, where $\lambda$ controls the adversarial strength. At equilibrium, $\mathcal{D}$ relies primarily on $\hat{Y}$ to predict $Y$, making $\phi(\mathbf{X})$ redundant given $\hat{Y}$ and thus approximating the desired conditional independence $Y \perp \phi(\mathbf{X}) \mid \hat{Y}$.
>
> * **Idea 3:** Theorem 4.2 suggests that maximizing conditional informativeness requires optimally assigning environment labels to data points. Although some prior work has explored environment-label assignment [1,2], the problem remains open under hidden-confounder shifts. Assigning environment labels based on latent confounder values may yield optimal performance. Once such environment labels are available, models such as mixtures of experts (e.g., [3]) and their variants can be used to learn environment-specific relationships.
>
> References:
>
> [1] Wu et al. Bridging multicalibration and out-of-distribution generalization beyond covariate shift. Advances in Neural Information Processing Systems, 2024.
>
> [2] Jiashuo et al. Data Heterogeneity Modeling for Trustworthy Machine Learning. arXiv preprint arXiv:2506.00969 (2025).
>
> [3] Prashant et al. Scalable out-of-distribution robustness in the presence of unobserved confounders. In The 28th International Conference on Artificial Intelligence and Statistics, 2025.

---

> ### Author Response · Authors · 2025-11-20
> **Response to Reviewer ov4h Continued**
>
> > Q2) I think Eq (2) and (3) should be derived formally in a proposition..
>
> Thank you for the suggestion. In the revised manuscript, we will move the formal details of the inequalities from the proof of Theorem 3.2 (currently in the Appendix) into the main text and present them as a standalone proposition.
>
>
> > Q3) Estimating MI of continuous, high-dimensional variables is a non-trivial problem....
>
> Thank you for the suggestion. We have introduced the details of MI estimation in Section 5 of the revised paper and referred to Appendix B for the details.
>
> > Q4) Presentation of results in Table 2: as is, it does not clearly show the correlation..
>
> We agree with you. In the revised version of the paper, we've mentioned the Spearman correlation coefficient $(\rho)$ in the text and removed the redundant table. We also include the values from Figure C4, which makes the estimation of $\rho$ more robust.
>
> > Q5) Proof of theorem 4.2 should be referred to in the main text.
>
> We have moved the proof to the main paper in the revised version of the paper.
>
> > Q6) An example graph similar to Figure 3 (or an extension thereof) would be helpful to understand definition 4.3 / the concept of informative covariates.
>
> Thank you for the suggestion. In the revised version of the paper, we have added a paragraph after Definition 4.3 describing the causal graph with informative covariates.
>
> > Q7) Line 76: Why is learning an invariant representation sufficient? The paper seems to suggest the opposite in Eq 1?
>
> Invariant representations are sufficient in the sense that, for the causal graph $E\leftrightarrow U; U\rightarrow \mathbf{X}; U\rightarrow Y; \mathbf{X}\leftrightarrow Y$ (shown in Figure 3), $\mathbb{P}(Y\mid \mathbf{X}, U)$ is invariant because conditioned on $U$, $Y$ is independent of $E$. That is, when $U$ is observed, using both $\mathbf{X}, U$ as inputs and employing invariance constraints on learning leads to generalization. However, since we do not observe $U$ in our setting, following Theorem 4.2, our theory suggest domain specific information.
>
> > Q8) Line 209: Why is the sum over $\mathcal{R}^{\mathcal{D}^e}$ and not simply over $\mathcal{R}^e$? For the other methods in the same paragraph the objective is always stated using $\mathcal{R}^e$.
>
> Thank you for the suggestion. We have replaced $\mathcal{R}^{\mathcal{D}^e}$ with $\mathcal{R}^{e}$ in Line 209 in the revised paper.
>
> We hope our responses have addressed your concerns. We are happy to answer any questions the reviewer may have.

---

> > ### Author Response · Authors · 2025-11-27
> > **Request for Feedback on Rebuttal**
> >
> > We would appreciate the reviewer’s feedback on our rebuttal. We believe we have addressed all raised concerns and are happy to answer any further questions.

---

### Official Review · Reviewer_ZFUv · 2025-10-30

**Soundness:** 3
**Presentation:** 3
**Contribution:** 4
**Rating:** 8
**Confidence:** 3

**Summary:**

This paper provides a comprehensive information-theoretic framework to understand various components of distribution shift in out-of-domain generalization. Based on existing puzzles on why simple ERM outperforms robust/causal methods, the authors point out -- through both careful theoretical analysis and empirical benchmark -- that confounding bias is the key issue that affects generalization performance, and adding non-causal yet confounding-informative variables can improve generalization. The proposed theory centers around a decomposition of the predictive information $I(Y;\hat{Y})$ into several terms related to environment change and variable-related shift terms.  This result is well explained through several results in representative settings. The proposed theory is corroborated via numerical experiments and provides insights on generalization performance in popular distribution shift benchmark datasets.

**Strengths:**

1. Understanding out-of-domain generalization is a very important problem, and the paper targets at a key issue in this area, which is the performance gap between robustness-oriented methods and plain ERM. The results in the paper should benefit the community in both understanding and subsequent method development.
2. The paper is well-written and well-structured, with rich discussion and inspiring insights.
3. The theory is supported by solid experiments and empirical insights.

**Weaknesses:**

1. Maybe I missed something but it would be helpful to clarify what datasets the results in Section 5 are from (are they from synthetic data or the TableShift benchmark?). Seems the results are from one single dataset? Or did you merge all the data?
2. While the authors clarify that this paper aims to provide empirical insights instead of solutions, which is totally fair, it might be useful to suggest some solutions based on the observed results.

**Questions:**

1. (same as weakness 1) How are the results in Section 5 obtained -- are they from a single dataset (synthetic or real) or merged everything?
2. The sign consistency metric is interesting. It's natural to expect a high fraction when a decomposition term is related to the performance. However, I'm a bit confused as to how to interpret a metric as low as $0.2$ (e.g., FS). Even for random guess (sth totally unrelated to prediction performance) the metric should be like $0.5$. Does a low metric mean this term predicts the improvement accurately in an opposite direction? If so, does it mean the theory is not accurate?
3. Given the important role of CI and CS in sign consistency metric, would you suggest optimizing them in training OOD generalization algorithms?
4. A potential use of the results is to estimate these terms in the existing environments to guide model choice. Could the authors comment on such applications, e.g., could it work and how future method development may involve them?

---

> ### Author Response · Authors · 2025-11-20
> **Response to Reviewer ZFUv**
>
> We thank you for encouraging and insightful review. Below we address each of your comments.
>
> > Q1) Maybe I missed something but it would be helpful ...?
>
> Results in Section 5 are based on both real-world and synthetic data. For real-world data results shown in Table 2 and Table 4, we show average results over eight datasets. Dataset specific results are presented in Appendix C. Results in Figure 4 correspond to the synthetic data.
>
> > Q2)  While the authors clarify that this paper aims to provide empirical insights instead of solutions, which is totally fair, it might be useful...
>
> We are glad to see your interest in further methodological developments based on our work. Below we provide few concrete ideas for actionable algorithms, each of which could motivate a separate line of research that is beyond the scope of this paper.
>
> **Potential ideas for actionable algorithms:**
>
> * **Idea 1:** Based on the synthetic experiments in Figure 2, adding domain-specific information $\mathbf{X}_I$ that is informative about $Y$ can improve model performance. From the causal graph in Figure 3, such domain-specific signals are likely associated with the hidden confounder. Adding these variables increases conditional informativeness and reduces concept shift. This suggests that $\mathbb{P}(Y \mid \mathbf{X}, \mathbf{X}_I)$ is closer to invariant than $\mathbb{P}(Y \mid \mathbf{X})$. Designing a systematic informative-covariate augmentation pipeline, either by collecting extra features or by generating them, would provide immediate practical benefits.
>
>
> * **Idea 2:** From Theorem 4.2, minimizing the residual can also improve performance. Making the residual term $I(\phi(\mathbf{X}); Y \mid \hat{Y})$ zero is equivalent to enforcing the conditional independence $Y \perp \phi(\mathbf{X}) \mid \hat{Y}$. One approach to achieve this is imposing an adversarial conditional-independence penalty. Specifically, define a base model $\mathcal{B}$ that learns $\phi(\mathbf{X})$ and predicts $\hat{Y} = f(\phi(\mathbf{X}))$ by minimizing the usual cross-entropy loss $L_{\text{base}}$. Introduce an adversarial model $\mathcal{D}$ that predicts $Y$ from $(\phi(\mathbf{X}), \hat{Y})$ using its own cross-entropy loss $L_{\text{adv}}$. Train the models adversarially: $\mathcal{D}$ minimizes $L_{\text{adv}}$, while $\mathcal{B}$ minimizes $L_{\text{base}} - \lambda L_{\text{adv}}$, where $\lambda$ controls the adversarial strength. At equilibrium, $\mathcal{D}$ relies primarily on $\hat{Y}$ to predict $Y$, making $\phi(\mathbf{X})$ redundant given $\hat{Y}$ and thus approximating the desired conditional independence $Y \perp \phi(\mathbf{X}) \mid \hat{Y}$.
>
> * **Idea 3:** Theorem 4.2 suggests that maximizing conditional informativeness requires optimally assigning environment labels to data points. Although some prior work has explored environment-label assignment [1,2], the problem remains open under hidden-confounder shifts. Assigning environment labels based on latent confounder values may yield optimal performance. Once such environment labels are available, models such as mixtures of experts (e.g., [3]) and their variants can be used to learn environment-specific relationships.
>
> References:
>
> [1] Wu et al. Bridging multicalibration and out-of-distribution generalization beyond covariate shift. Advances in Neural Information Processing Systems, 2024.
>
> [2] Jiashuo et al. Data Heterogeneity Modeling for Trustworthy Machine Learning. arXiv preprint arXiv:2506.00969 (2025).
>
> [3] Prashant et al. Scalable out-of-distribution robustness in the presence of unobserved confounders. In The 28th International Conference on Artificial Intelligence and Statistics, 2025.

---

> ### Author Response · Authors · 2025-11-20
> **Response to Reviewer ZFUv Continued.**
>
> > Q3) The sign consistency metric is interesting. It's natural to expect a high fraction when a decomposition term is related to the performance. However, I'm a bit confused as to how to interpret a metric as low as 0.2 (e.g., FS). Even for random guess (sth totally unrelated to prediction performance) the metric should be like 0.5. Does a low metric mean this term predicts the improvement accurately in an opposite direction? If so, does it mean the theory is not accurate?
>
> The sign-consistency metric measures whether a technique/idea (e.g., adding informative covariates) moves the terms in Equation (1) in a particular direction. It is not necessary for a technique/idea to achieve ideal sign-consistency for every term.
>
> Following the similar proof techniques used in Proposition 4.1, we can show that the residual term may increase after adding informative covariates. This is reflected in Table 2, where sign-consistency values for residual-related terms are below 0.5.
>
> The synthetic-data results in Figure 4 show the following trends: after adding informative covariates, conditional informativeness (CI) and feature shift (FS) increase, concept shift (CS) decreases, and mean squared error (MSE) is reduced. Variation (Var), however, remains near zero in these experiments and thus shows no measurable effect on performance. These observations are consistent with the Proposition. On real-world datasets, we also observe generally high sign-consistency values for CI and CS after adding informative covariates. Note that a high sign-consistency value for every term is not required for good generalization: for instance, a model that explicitly penalizes feature shift may produce a low sign-consistency for FS, and a model that does not explicitly minimize residuals may show low sign-consistency for residual-related terms.
>
> Yet, across experiments, CI and CS show the strongest and most consistent association with performance gains, as measured by the sign-consistency metric. An advantage of the sign-consistency metric is that stable patterns such as these can help guide the design of downstream algorithms tailored to each dataset.

---

> > ### Author Response · Authors · 2025-11-27
> > **Request for Feedback on Rebuttal**
> >
> > We would appreciate the reviewer’s feedback on our rebuttal. We believe we have addressed all raised concerns and are happy to answer any further questions.

---

### Official Review · Reviewer_cZis · 2025-11-05

**Soundness:** 3
**Presentation:** 4
**Contribution:** 3
**Rating:** 6
**Confidence:** 3

**Summary:**

The paper provides a theoretical and empirical explanation for observed paradoxes in Out-of-Distribution (OOD) generalization, specifically why conventional Empirical Risk Minimization (ERM) models often match or surpass state-of-the-art OOD methods, and why including all available covariates, even non-causal ones, improves OOD accuracy. The core argument states that shifts in hidden confounding variables induce simultaneous distribution shifts that invalidate the assumptions of invariance-focused OOD methods. Under this causal structure, the authors prove that generalization success depends on learning environment-specific relationships. The benefit of non-causal but informative covariates is explained by their ability to act as proxies for the hidden confounders, thereby mitigating the negative effects of the shifts and increasing conditional informativeness. This conditional informativeness measure correlates strongly with both In-Distribution and OOD accuracy across eight real-world datasets, validating the hypothesis that under hidden confounding, exploiting environment-specific information is necessary for robustness.

**Strengths:**

The work theoretically justifies the importance of non-causal, informative covariates (XI) by showing they help maximize generalization performance.  Proposition 4.1 demonstrates that adding these variables reduces concept shift and increases conditional informativeness, thereby mitigating the negative impact of unobserved confounders.



Experiments using synthetic data with known causal structure and extensive testing on eight real-world tabular datasets (TableShift benchmark) consistently confirm the core hypotheses: hidden confounding is prevalent (inducing simultaneous shifts), and conditional informativeness is highly correlated with both ID and OOD accuracy.

**Weaknesses:**

The core theoretical insights (Theorems 4.2 and related decompositions) are dependent on assuming a very specific, unobserved causal graph. This foundational assumption remains unverifiable in real-world data. This limits the ability of the derived guidance to be applied with certainty, as the true underlying causal structure is unknown.

The empirical validation is focused almost exclusively on tabular prediction tasks employing relatively simple models like XGBoost and MLP. The role of the feature extractor in complex domains is much more sophisticated. It is questionable whether the conclusion that invariance is insufficient and environment-specific information should be prioritized holds true when the extractor is a large, non-linear model and the data shifts are visual or semantic rather than structural/tabular.

The authors explicitly state that their primary goal is to explain existing phenomena rather than provide a novel solution to the problem of hidden confounding shift. While the theoretical explanation is robust, the paper primarily offers a lens through which to understand why simple methods already work well, rather than proposing a new method that reliably and optimally maximizes conditional informativeness across generalized shift settings.

The identification of "informative covariates" is critical to the proposed strategy. However, outside of carefully constructed synthetic settings, rigorously identifying these non-causal proxy variables remains a challenge. The reliance on LLM queries for semantic insight into potential confounders in real-world datasets underscores the difficulty of establishing these variables non-qualitatively.

**Questions:**

Could the authors elaborate on the architectural properties of XGBoost that make it inherently proficient at capturing the environment-specific relationships required for maximizing conditional informativeness compared to IRM, which attempts to enforce invariance by minimizing penalties associated with environment dependency?

Given the complexity of feature extraction in computer vision, how would the predictive information decomposition terms (especially variation and feature shift behave in a system dealing with spurious correlations in image data, and what theoretical adjustments might be needed to apply this framework to deep OOD models outside the tabular setting?

---

> ### Author Response · Authors · 2025-11-20
> **Response to Reviewer cZis**
>
> We thank you for your insightful review. We will address each of your comments below.
>
> > Q1) The core theoretical insights (Theorems 4.2 and related decompositions) are dependent on assuming a very specific, unobserved causal graph. This foundational assumption remains unverifiable in real-world data. This limits the ability of the derived guidance to be applied with certainty, as the true underlying causal structure is unknown.
>
> The causal graph used in this work is intentionally simple and broadly applicable. Although its assumptions are not always directly verifiable, many real-world data-generating processes can be modeled by this graph.
> As discussed in Lines 321--323 of the main paper, the causal direction $\mathbf{X}\to Y$ is very common in tabular datasets [1]. Moreover, any shift in the joint distribution $\mathbb{P}(\mathbf{X},Y)$ can be represented as a confounding shift (Figure 3). The notion of informative covariates is defined via a conditional-independence statement (Definition 4.3) but not based on a causal structure. Thus, the simple structure among $\mathbf{X},Y,U$ captures many practical settings, and our empirical results align with the theoretical predictions in many real-world datasets (see Table 4), supporting the prevalence of the considered causal model in real-world data.
>
> References:
>
> [1] Nastl, Vivian, and Moritz Hardt. "Do causal predictors generalize better to new domains?." Advances in Neural Information Processing Systems 37 (2024): 31202-31315.
>
> > Q2) The empirical validation is focused almost exclusively on tabular prediction tasks employing relatively simple models like XGBoost and MLP. The role of the feature extractor in complex domains is much more sophisticated. It is questionable whether the conclusion that invariance is insufficient and environment-specific information should be prioritized holds true when the extractor is a large, non-linear model and the data shifts are visual or semantic rather than structural/tabular.
>
> In vision datasets, the input often contains most of the information needed to predict the label, because labels are typically assigned by experts directly from the image. Consequently, vision research usually focuses on identifying causal visual features versus background or noisy features and on making models robust to common image shifts (for example, changes in background, rotation, or lighting) that can confound inputs and labels. In the context of our causal graph (Figure 3), these efforts target causal-feature discovery and robust representation learning rather than the addition of external informative covariates.
>
> By contrast, tabular datasets more frequently suffer from missing causal or informative covariates and from hidden confounding (see Section 2, Paragraph 2 in [1]). When everything required to predict $Y$ is already present in the inputs, invariance techniques combined with effective representation extractors may suffice for good generalization. Because tabular data often lack such completeness, we focus on tabular settings, which pose distinct and practically important challenges.
>
> Even when all causal features are available (like in typical vision problems), we agree that applying our information-theoretic decomposition is an interesting direction for future work. In particular, identifying the operating environment and exploiting appropriate inductive biases (for example, the prior that a fish is more likely to appear in water than in grass) could improve performance. Studying how these ideas interact with our decomposition could yield useful insights.
>
>
> References:
>
> [1] Liu, Jiashuo, et al. "On the need for a language describing distribution shifts: Illustrations on tabular datasets." Advances in Neural Information Processing Systems, 2023.
>
> > Q3) The identification of "informative covariates" is critical to the proposed strategy. However, outside of carefully constructed synthetic settings, rigorously identifying these non-causal proxy variables remains a challenge. The reliance on LLM queries for semantic insight into potential confounders in real-world datasets underscores the difficulty of establishing these variables non-qualitatively.
>
> In tabular data, features associated with the target are more likely to be informative covariates: hidden confounders—common in real-world datasets—can causally affect both the target and such features. Hence, any covariate that is associated with target is most likely to be informative. There exists cases where collecting additional informative covariates can be difficult in practice. When this is the case, one can sometimes construct synthetic covariates from original data as explained in Figure 2 subplot 2.

---

> ### Author Response · Authors · 2025-11-20
> **Response to Reviewer cZis Continued.**
>
> > Q4) The authors explicitly state that their primary goal is to explain existing phenomena rather than provide a novel solution to the problem of hidden confounding shift. While the theoretical explanation is robust, the paper primarily offers a lens through which to understand why simple methods already work well, rather than proposing a new method that reliably and optimally maximizes conditional informativeness across generalized shift settings.
>
>
> We note that the main goal of this work is to provide theoretical and empirical explanations for the observed phenomena in the literature: (i) ERM can outperform OOD generalization methods, and (ii) informative but non-causal covariates can improve OOD generalization performance. In this process, we proved results showing that conditional informativeness and the residual play a crucial role in generalization performance, and that informative covariates influence the various factors in Equation (1) responsible for OOD generalization. These results give rise to the following ideas, each of which could motivate a separate line of research that is beyond the scope of this paper.
>
> **Potential ideas for actionable algorithms:**
>
> * **Idea 1:** Based on the synthetic experiments in Figure 2, adding domain-specific information $\mathbf{X}_I$ that is informative about $Y$ can improve model performance. From the causal graph in Figure 3, such domain-specific signals are likely associated with the hidden confounder. Adding these variables increases conditional informativeness and reduces concept shift. This suggests that $\mathbb{P}(Y \mid \mathbf{X}, \mathbf{X}_I)$ is closer to invariant than $\mathbb{P}(Y \mid \mathbf{X})$. Designing a systematic informative-covariate augmentation pipeline, either by collecting extra features or by generating them, would provide immediate practical benefits.
>
>
> * **Idea 2:** From Theorem 4.2, minimizing the residual can also improve performance. Making the residual term $I(\phi(\mathbf{X}); Y \mid \hat{Y})$ zero is equivalent to enforcing the conditional independence $Y \perp \phi(\mathbf{X}) \mid \hat{Y}$. One approach to achieve this is imposing an adversarial conditional-independence penalty. Specifically, define a base model $\mathcal{B}$ that learns $\phi(\mathbf{X})$ and predicts $\hat{Y} = f(\phi(\mathbf{X}))$ by minimizing the usual cross-entropy loss $L_{\text{base}}$. Introduce an adversarial model $\mathcal{D}$ that predicts $Y$ from $(\phi(\mathbf{X}), \hat{Y})$ using its own cross-entropy loss $L_{\text{adv}}$. Train the models adversarially: $\mathcal{D}$ minimizes $L_{\text{adv}}$, while $\mathcal{B}$ minimizes $L_{\text{base}} - \lambda L_{\text{adv}}$, where $\lambda$ controls the adversarial strength. At equilibrium, $\mathcal{D}$ relies primarily on $\hat{Y}$ to predict $Y$, making $\phi(\mathbf{X})$ redundant given $\hat{Y}$ and thus approximating the desired conditional independence $Y \perp \phi(\mathbf{X}) \mid \hat{Y}$.
>
> * **Idea 3:** Theorem 4.2 suggests that maximizing conditional informativeness requires optimally assigning environment labels to data points. Although some prior work has explored environment-label assignment [1,2], the problem remains open under hidden-confounder shifts. Assigning environment labels based on latent confounder values may yield optimal performance. Once such environment labels are available, models such as mixtures of experts (e.g., [3]) and their variants can be used to learn environment-specific relationships.
>
> References:
>
> [1] Wu et al. Bridging multicalibration and out-of-distribution generalization beyond covariate shift. Advances in Neural Information Processing Systems, 2024.
>
> [2] Jiashuo et al. Data Heterogeneity Modeling for Trustworthy Machine Learning. arXiv preprint arXiv:2506.00969 (2025).
>
> [3] Prashant et al. Scalable out-of-distribution robustness in the presence of unobserved confounders. In The 28th International Conference on Artificial Intelligence and Statistics, 2025.

---

> ### Author Response · Authors · 2025-11-20
> **Response to Reviewer cZis Continued.**
>
> > Q5) Could the authors elaborate on the architectural properties of XGBoost that make it inherently proficient at capturing the environment-specific relationships required for maximizing conditional informativeness compared to IRM, which attempts to enforce invariance by minimizing penalties associated with environment dependency?
>
> XGBoost captures environment-specific patterns through its tree-based structure. Individual decision trees partition the input space into regions where label behavior can differ, so the ensemble effectively learns distinct rules for different regions. Viewed as an ensemble, XGBoost behaves like a weighted mixture-of-experts: individual trees specialize on particular input regions (which can be interpreted as latent environments), automatically grouping data points into adaptive clusters. Unlike methods such as IRM that require predefined environment labels $E$ (for example, assigning all data from a geographic region to one environment), XGBoost induces its partitions directly from the data, often producing more flexible groupings than fixed, externally provided labels. This intrinsic partitioning and specialization helps explain XGBoost’s empirical success under hidden-confounding shifts and aligns with the findings in [1]. While some invariant methods suppress environment-specific signals, XGBoost can exploit those signals by first identifying local contexts and then maximizing predictive accuracy within each context.
>
> A complementary view focuses on boosting dynamics. At each iteration, gradient-boosting fits the residuals $Y-\hat{Y}$, thereby emphasizing examples the current model predicts poorly. If these residuals correlate with latent environment partitions, the reweighting/focusing behavior causes subsequent trees to learn different rules for different partitions.
>
> References:
>
> [1] Popov, Sergei, Stanislav Morozov, and Artem Babenko. "Neural Oblivious Decision Ensembles for Deep Learning on Tabular Data." International Conference on Learning Representations. 2020
>
> > Q6) Given the complexity of feature extraction in computer vision, how would the predictive information decomposition terms (especially variation and feature shift behave in a system dealing with spurious correlations in image data, and what theoretical adjustments might be needed to apply this framework to deep OOD models outside the tabular setting?
>
>
> In computer vision tasks, a causal graph similar to the one in Figure 3 can be considered. Feature shift usually happens between environments (e.g., one environment contains photographed animal images while another contains painted or sketched animal images). Similarly, one may observe variation for a fixed label (e.g., two different environments with images of two different dog breeds). Feature shift is difficult to avoid because the OOD nature of the data allows feature distributions to change across environments.
>
> To minimize the variation $I(\phi(\mathbf{X}); E \mid Y)$, counterfactual data augmentation (e.g., [1]) has been shown to be an effective approach. Specifically, for a given $Y$ (e.g., a dog image), counterfactual augmentation methods typically intervene on the environment variable (e.g., background) and generate new images with the same label and the same causal features (e.g., counterfactual dog images with different backgrounds with same causal features). Such counterfactual data help remove spurious associations between causal features $\mathbf{X}$, the label $Y$, and the environment variable $E$. Because counterfactual generation in image datasets is challenging, we believe research on removing spurious correlations focuses more on effective feature extraction and composition to generate realistic counterfactual images.
>
> References:
>
> [1] Sauer, Axel, and Andreas Geiger. "Counterfactual Generative Networks." International Conference on Learning Representations. 2021.

---

> > ### Author Response · Authors · 2025-11-27
> > **Request for Feedback on Rebuttal**
> >
> > We would appreciate the reviewer’s feedback on our rebuttal. We believe we have addressed all raised concerns and are happy to answer any further questions.

---

### Official Review · Reviewer_5zx4 · 2025-11-05

**Soundness:** 3
**Presentation:** 3
**Contribution:** 2
**Rating:** 4
**Confidence:** 3

**Summary:**

The established understanding of OOD generalization suggests that learning invariant and causal representations improves generalization. However, recent empirical work has shown two things: 1)ERM often generalizes better than methods designed for OOD generalization 2)Non-causal features can improve performance. This paper gives theoretical justification for these empirical findings. This work shows that under hidden-confounder shift, invariant causal representations could be suboptimal to ERM and using non-causal features. They also empirically verify that several real datasets do have hidden confounder shifts and their OOD performance using various approaches can be explained by this work’s theoretical framework.

**Strengths:**

This paper provides a neat information theory based framework to understand the OOD performance of predictors under hidden confounding shift
The theory explains two empirical phenomenon: 1) Learning invariant representations is not optimal for hidden confounder shift 2)Non-causal features can improve performance
The theory does indeed explain the empirical OOD performance of different training algorithms and the different sets of features (causal, arguably causal, all) used. However, in practice E is often hidden and calculating the terms would not be possible.
The paper is generally well-written

**Weaknesses:**

Regarding invariant representations: It is well known that under hidden confounder shift, they would underperform compared to methods that use information about the environment or its proxies. This intuitively makes sense since having information about the environment should help over methods that do not consider that. I believe the primary strength of invariant methods is in cases where the environment in test is out of the support of the train environments. Therefore, I do not see explaining the underperformance of invariant representations in hidden confounder shift setting a substantial contribution.
Regarding using non-causal features improves performance: The way I intuitively understand this is that if there is a hidden confounder, then we do not observe all causal variables. Therefore, conditioning on the observed causal variables does not d-separate the non-causal ones. Hence, using non-causal features may help. It's unclear what additional insight the information-theoretic framework provides. Moreover, the d-separation argument applies to other hidden variable scenarios as well (e.g., measurement bias).

**Questions:**

Why is it surprising/insightful that environment information should give better models? If not, are there any quantitative predictions we can make in practice when E is hidden?

Why extra insight does the information theory framework provide that the d-separation argument does not?
Open to raising the rating if questions are addressed.

---

> ### Author Response · Authors · 2025-11-20
> **Response to Reviewer 5zx4**
>
> Thank you for your insightful review. Below we address each of your comments.
>
> > Q1) Regarding invariant representations: It is well known... This intuitively makes sense since... I believe the primary strength of invariant methods is in cases... Therefore, I do not see explaining...
>
> Our work is the first to provide formal theoretical justification, going beyond intuition, for two observed phenomena: (i) ERM performs better than specialized methods for OOD generalization, (ii) non causal but informative covariates improves performance. At one extreme, several empirical studies show that invariant learning methods can be outperformed by ERM-based models without offering theoretical explanations. In fact, the research community has been surprised by the empirical observation that ERM relying on more (potentially non-causal) variables ranks among the best OOD robust methods [1,2]. At the other extreme, recent work has proposed methods that directly address hidden confounding shifts (but missing the potential of using informative but non-causal features). Thus, a clear research gap exists between empirical findings and recent methodological developments. We bridge that gap by showing a) that common OOD benchmarks are governed by hidden confounding shift, b) how hidden confounding shift motivates domain-specific predictors, and c) how informative covariates facilitate OOD generalization.
>
> Specifically, we make the following novel contributions:
> We show that variation, concept shift, label shift, and feature shift terms tend to cancel out so that the only two dominant factors affecting OOD generalization are conditional informativeness and the residual (Theorem 4.2). Addressing hidden-confounding shifts requires extra assumptions about the data-generating process (e.g., access to proxy variables [3]), which may be infeasible in many applications. To assess this, we provide a diagnostic tool for practitioners to assess how strongly a dataset is affected by hidden-confounding shifts.
>
> References:
> [1] Nastl et al. Do causal predictors generalize better to new domains?. Advances in Neural Information Processing Systems, 2024.
>
> [2] Joshua et al. Benchmarking distribution shift in tabular data with tableshift. In Thirty-seventh Conference on Neural Information Processing Systems
> Datasets and Benchmarks Track, 2023.
>
> [3] Prashant et al. Scalable out-of-distribution robustness in the presence of unobserved confounders. In The 28th International Conference on Artificial Intelligence and Statistics, 2025.
>
> > Q2) Regarding using non-causal features improves performance: The way I intuitively understand this is that if there is a hidden confounder, then we do not observe all causal variables. Therefore, conditioning on the observed causal variables does not d-separate the non-causal ones. Hence, using non-causal features may help. It's unclear what additional insight...
>
> We note that non-causal covariates improve performance only when they are informative. If non-causal covariates are not informative, relying on them causes the model to overfit the training data. Recent work [1] finds the performance gains from adding possibly non-causal covariates surprising, since conventional wisdom suggests that only causal covariates guarantee OOD generalization. The authors of [1] also emphasize the need for a theoretical understanding of how non-causal covariates influence generalization. In this work, in addition to the notions of causal and non-causal covariates, we introduce the distinction between informative and non-informative covariates. Further, [2] argues that adding features that reduce the $Y\mid\mathbf{X}$ shift leads to improved performance. In our framework, such a reduction of the $Y\mid\mathbf{X}$ shift can be interpreted as a reduction in concept shift.
>
> d-separation identifies which covariates are informative, but it does not quantify their impact. Our proposed information-theoretic framework allows us to characterize how these informative covariates influence the terms in Equation (1), providing a more principled understanding of their role in OOD generalization (Proposition 4.1).
>
> References:
> [1] Nastl et al. Do causal predictors generalize better to new domains?. Advances in Neural Information Processing Systems, 2024.
>
> [2] Liu et al. On the need for a language describing distribution shifts: Illustrations on tabular datasets. Advances in Neural Information Processing Systems, 2023.

---

> ### Author Response · Authors · 2025-11-20
> **Response to Reviewer 5zx4 continued**
>
> > Q3) Why is it surprising/insightful that environment information should give better models? If not, are there any quantitative predictions we can make in practice when E is hidden?
>
> Not every environment-specific feature is useful. Some may cause the model to overfit the training data. Only those features $\mathbf{X}_I$ for which $\mathbb{P}(Y \mid \mathbf{X}, \mathbf{X}_I)$ is approximately invariant across environments are helpful. (Recall that $\mathbb{P}(Y \mid \mathbf{X})$ is not invariant, whereas $\mathbb{P}(Y \mid \mathbf{X}, U)$ is, according to the causal graph in Figure 3.) In other words, environment-specific, informative features are useful only if they provide information about the hidden confounder.
>
> In practice, $E$ is not always observed. However, many high-performing models such as XGBoost do not require $E$ explicitly. While some methods (such as invariant learning) require environment labels, XGBoost can implicitly learn adaptive clusters via its decision trees: it effectively identifies local contexts and maximizes predictive accuracy within each one. Therefore, to improve generalization it is crucial to supply additional information that increases conditional informativeness — for example, informative covariates (Proposition 4.1) or explicit regularization that promotes conditional informativeness. These techniques help XGBoost learn about $E$ implicitly, leading to better OOD performance.
>
> > Q4) Why extra insight does the information theory framework provide that the d-separation argument does not? Open to raising the rating if questions are addressed.
>
> The d-separation argument is used only to formalize the notion of informative covariates in Section 4.3. d-separation criteria do not explain how informative covariates affect generalization. However, our information-theoretic framework shows how the terms in Equation (1) alter with the addition of informative covariates. In particular, addition of informative covariates reduces concept shift, increases conditional informativeness, etc., as shown in Proposition 4.1. Information theoretic terms and their interaction in different settings are more general than the d-separation criteria. Any algorithmic idea (e.g., informative feature augmentation or invariance regularization) can be analyzed by understanding the behaviour of the terms in Equation (1).
>
> We hope our responses have addressed your concerns. We are happy to answer any questions you may have.

---

> > ### Author Response · Authors · 2025-11-27
> > **Request for Feedback on Rebuttal**
> >
> > We would appreciate the reviewer’s feedback on our rebuttal. We believe we have addressed all raised concerns and are happy to answer any further questions.

---

### Author Response · Authors · 2025-11-20
**Common Response to All Reviewers**

We thank all the reviewers for their insightful and encouraging reviews. In particular, reviewers noted that (i) the paper is well written (5zx4, ZFUv), (ii) the research problem is timely and relevant to the community (ZFUv, ov4h), (iii) the theoretical contributions are novel and useful (ov4h), and (iv) the experiments are extensive and support the theory (cZis, ZFUv, ov4h). We will address each reviewer’s comments individually below.

---

### Author Response · Authors · 2025-12-03
**Summary Comment to Area Chair**

Dear Area Chair,

We thank you for your time and effort in reviewing our paper and the rebuttal. Below, we provide a summary of the paper, the reviews and our rebuttal.

**Paper summary:** In this paper, our goal is to explain two empirical observations that the research community found surprising: (i) ERM-based models tend to outperform specialized methods for OOD generalization such as invariance-learning and robust-learning approaches, and (ii) the addition of informative but non-causal covariates improves OOD generalization performance (for most models). To the best of our knowledge, we are the first to provide a theoretical explanation (corroborated by experimental evidence) for these recent empirical observations found in the literature. To this end, we introduce two novel information-theoretic decompositions of predictive information that apply to general distribution shift and to distribution shift induced by hidden confounding (Equations 1 and 4). Our empirical results demonstrate the prevalence of hidden confounding shift in real-world data. Under hidden confounding shift, we theoretically show that the best generalization performance is achieved by maximizing the quantity conditional informativeness (CI) – residual (RES). Our experiments on real-world datasets confirm our theory: CI – RES correlates strongly with both in-distribution (ID) and out-of-distribution (OOD) accuracy, with the highest accuracy achieved by ERM-based models such as MLP and XGBoost. The experiments also show that adding informative but non-causal covariates increases conditional informativeness, thereby improving generalization—providing both theoretical and empirical justification for this phenomenon.

**Summary of reviews:** Reviewers recognized the significance of the research problem and the novelty of our contributions. In particular, reviewers noted that the paper is well written (5zx4, ZFUv), the research problem is timely and relevant to the community (ZFUv, ov4h), the theoretical contributions are novel and useful (ov4h), and the experiments are extensive and support the theory (cZis, ZFUv, ov4h). No reviewer raised critical concerns about the paper’s core validity, and reviewers assessed the paper as high-quality with an **average score of $6$ before rebuttal**. Most reviewer comments concerned (i) clarifications, (ii) editorial improvements, (iii) applicability of our theory to other modalities such as images, and (iv) potential actionable algorithms derived from our framework.

**Author rebuttal:**  We addressed all reviewer comments thoroughly, including references, examples, and concrete ideas for actionable algorithms for future research motivated by our theory. Reviewer 5zx4 raised questions about the novelty of our paper by asking the questions: (i) what are the uses of information theoretic framework and (ii) what are the uses of environment specific covariates. These are about clarifications of our work and our responses clearly answered their questions. **Reviewer 5zx4 also indicated their willingness to increase their score** if our responses address their concerns.

We have updated the paper based on our rebuttal and changes compared to the original submission are minimal.

We appreciate your time and effort.

---

### Meta-Review · Area_Chair_2u8e · 2026-01-07

**Summary:**

This paper provides a novel information-theoretic framework to explain why ERM and non-causal covariates often outperform specialized OOD methods. By attributing these paradoxes to hidden confounding shifts, the authors prove that environment-specific relationships are essential for generalization when invariant assumptions are violated. The theory is well-supported by synthetic experiments and the TableShift benchmark.

**Reviewer Concerns:**

Technical queries regarding algorithmic actionability, domain scope (tabular vs. vision), and MI estimation were satisfactorily addressed through the authors' detailed rebuttal and manuscript revisions.

**Reviewer Scores:**

With scores of 8, 6, 6, and 4, the consensus is positive, highlighting the paper's timely contribution and strong alignment between theory and empirical results.

---

### Decision · Program_Chairs · 2026-01-26

Accept (Poster)